# Non-rapid eye movement sleep and wake neurophysiology in schizophrenia

Nataliia Kozhemiako[1†], Jun Wang[2†], Chenguang Jiang[2†], Lei A Wang[3], Guanchen Gai[2], Kai Zou[2], Zhe Wang[2], Xiaoman Yu[2], Lin Zhou[3], Shen Li[4], Zhenglin Guo[3], Robert Law[1], James Coleman[3], Dimitrios Mylonas[5], Lu Shen[6], Guoqiang Wang[2], Shuping Tan[7], Shengying Qin[6], Hailiang Huang[3,8], Michael Murphy[4], Robert Stickgold[9,10], Dara Manoach[5], Zhenhe Zhou[2*‡], Wei Zhu[2‡], Mei-Hua Hal[4*‡], Shaun M Purcell[1,10*‡], Jen Q Pan[3*‡]

[1]Department of Psychiatry, Brigham and Women's Hospital, Harvard Medical School, Boston, United States; [2]The Affiliated Wuxi Mental Health Center of Nanjing Medical University, Wuxi, China; [3]Stanley Center for Psychiatric Research, Broad Institute of MIT and Harvard, Cambridge, United States; [4]Department of Psychiatry, McLean Hospital, Harvard Medical School, Belmont, United States; [5]Department of Psychiatry, Massachusetts General Hospital, Harvard Medical School, Boston, United States; [6]Bio-X Institutes, Shanghai Jiao Tong University, Shanghai, China; [7]Huilong Guan Hospital, Beijing University, Beijing, China; [8]Analytic and Translational Genetics Unit, Massachusetts General Hospital, Harvard Medical School, Boston, United States; [9]Beth Israel Deaconess Medical Center, Boston, United States; [10]Department of Psychiatry, Harvard Medical School, Boston, United States

*For correspondence:
zhouzh@njmu.edu.cn (ZZ);
mhall@mclean.harvard.edu (M-HH);
smpurcell@bwh.harvard.edu (SMP);
jpan@broadinstitute.org (JQP)

†These authors are co-first authors to this work
‡These authors are co-senior authors to this work

Competing interest: The authors declare that no competing interests exist.

**Abstract** Motivated by the potential of objective neurophysiological markers to index thalamocortical function in patients with severe psychiatric illnesses, we comprehensively characterized key non-rapid eye movement (NREM) sleep parameters across multiple domains, their interdependencies, and their relationship to waking event-related potentials and symptom severity. In 72 schizophrenia (SCZ) patients and 58 controls, we confirmed a marked reduction in sleep spindle density in SCZ and extended these findings to show that fast and slow spindle properties were largely uncorrelated. We also describe a novel measure of slow oscillation and spindle interaction that was attenuated in SCZ. The main sleep findings were replicated in a demographically distinct sample, and a joint model, based on multiple NREM components, statistically predicted disease status in the replication cohort. Although also altered in patients, auditory event-related potentials elicited during wake were unrelated to NREM metrics. Consistent with a growing literature implicating thalamocortical dysfunction in SCZ, our characterization identifies independent NREM and wake EEG biomarkers that may index distinct aspects of SCZ pathophysiology and point to multiple neural mechanisms underlying disease heterogeneity. This study lays the groundwork for evaluating these neurophysiological markers, individually or in combination, to guide efforts at treatment and prevention as well as identifying individuals most likely to benefit from specific interventions.

## Editor's evaluation

This study, one of the largest of its kind, replicates previous findings regarding the impairment of sleep rhythms in patients with schizophrenia relative to healthy controls. Specifically, sleep spindles, which constitute a hallmark of non-Rapid Eye Movement sleep, are less frequent in people with schizophrenia and several other sleep features were also affected. Overall, this study provides

evidence that brain dynamics during sleep are promising biomarkers for the diagnosis and the prevention of schizophrenia.

## Introduction

Schizophrenia (SCZ) is a chronic disorder characterized by cognitive and behavioral dysfunction that significantly impacts the quality of life of affected individuals and their caregivers (*Charlson et al., 2018*; *Stanley et al., 2017*). It is highly heritable, exhibiting a heterogenous and polygenic architecture implicating many genetic risk factors (*Ripke, 2014*). Despite intensive research, current medications ameliorate positive symptoms only in a subset of individuals, often with side effects and with little impact on negative symptoms or cognitive deficits. Consequently, recovery outcomes have not improved over the past decades (*Jääskeläinen et al., 2013*). Given the clinical and genetic heterogeneity of SCZ, identifying reliable, objective biomarkers that index specific neurobiological deficits is crucial for developing the next-generation therapeutics.

Emerging evidence points to thalamus as a critical node that supports cognitive function, and to abnormal thalamocortical connectivity as a key neurobiological deficit in SCZ (*Woodward et al., 2012*; *Anticevic et al., 2015*; *Cao et al., 2018*; *Woodward and Heckers, 2016*; *Wu et al., 2022*). NREM (non-rapid eye movement) sleep offers a lens through which we may index thalamocortical function without confounds from waking behaviors such as active symptoms or altered motivation. Two hallmarks of NREM sleep measured by the electroencephalogram (EEG) – slow oscillations (SO) and spindles – reflect distinct thalamic and thalamocortical circuits. The slow (~1 Hz) neuronal oscillations with large amplitude are generated by cortical neurons and propagated by cortico-thalamo-cortical circuits, while spindles are bursts of oscillatory neural activity (typically between 10 and 16 Hz and ~1 s in duration) arising from reverberant interaction between thalamic reticular nucleus (TRN) and thalamocortical relay neurons and modulated by thalamocortical connections. The coupling of SOs and spindles mediates information transfer and storage during sleep that underpins NREM's role in overnight memory consolidation (*Walker and Stickgold, 2004*). NREM traits show strong heritability in healthy populations (*Ambrosius et al., 2008*), correlate with cognitive performances (*Geiger et al., 2011*; *De Gennaro et al., 2008*; *Purcell et al., 2017*), and afford objective, quantifiable markers of thalamic and thalamocortical functioning in large cohorts.

Impaired sleep spindles and SOs have been reported in SCZ patients. Despite relatively consistent support for a general reduction in spindle density and/or amplitude in SCZ patients (*Ferrarelli et al., 2007*; *Ferrarelli et al., 2010*; *Göder et al., 2015*; *Mylonas et al., 2020a*; *Schilling et al., 2017*; *Wamsley et al., 2012*), it is not clear how specific topographical and temporal spindle properties, which may reflect distinct thalamocortical connections, are associated with SCZ. For example, distinguishing fast and slow spindles (FS and SS) and their temporal coupling with SO are less studied in SCZ patients. More generally, a comprehensive examination of how NREM features (including existing and novel metrics) are altered in SCZ and how they relate to each other is not established (*Castelnovo et al., 2018*; *Zhang et al., 2020*). This is in part due to the relatively small sample sizes ($N < 30$) previously utilized, that were unable to support comprehensive analyses across multiple domains of NREM sleep.

More importantly, whether and how distinct NREM sleep deficits track with waking EEG, clinical symptoms, medication, and cognition within patients is not clear (*Au and Harvey, 2020*). For example, one recent report suggested reductions in spindle density were significantly more extensive in patients experiencing auditory hallucinations compared to hallucination-free patients (*Sun et al., 2021*). Such a link is intriguing, considering the substantial evidence of altered auditory processing in SCZ (*Erickson et al., 2016*; *Freedman et al., 2020*; *Thuné et al., 2016*), and that both spindles and auditory processing heavily rely on uninterrupted and precise function of thalamocortical circuits (*Ferrarelli and Tononi, 2010*). However, whether auditory abnormalities during wake and spindle deficits during sleep reflect similar or distinct thalamocortical dysfunctions in SCZ are yet to be investigated.

Here, using whole-night high-density electroencephalography (hdEEG), we comprehensively characterized alterations across multiple domains of NREM neurophysiology, their interdependencies and relationship to wake and clinical features in a new SCZ cohort of 72 patients and 58 healthy controls. We first sought to replicate the primary spindle density deficit and characterize the precise

facets of spindle activity associated with SCZ, including topography, morphology, and novel metrics of intra-spindle frequency modulation. Second, we extend our analyses to broader NREM sleep EEG, including spectral features, functional connectivity, SOs, and their coupling with spindles, to determine which features track with disease state, and whether their association with SCZ is independent of spindle parameters. Third, we asked whether spindle deficits likely index the same thalamocortical dysfunctions reflected in auditory processing abnormalities during wake, by determining how sleep and wake metrics were correlated within individuals. Fourth, we investigated the relationship between disease-associated metrics and symptom severity within patients. Fifth, we attempted to replicate our sleep EEG findings by compiling an independent and demographically distinct replication dataset of 57 cases and 59 controls and applying the same analytic procedures. Finally, based on multiple domains of EEG metric, we constructed joint models to classify diagnostic status and assessed their predictive performance in the original sample, as well as their transferability to the independent collection of patients with distinct demographics.

## Results

Whole-night sleep EEG and wake auditory event-related potential (ERP) data were acquired from 130 individuals – 72 patients diagnosed with SCZ (25 females, 34.8 ± 7.0 years of age) and 58 healthy volunteers (CTR, 25 females, 31.7 ± 6.3 years of age) with no personal or family history of SCZ spectrum disorders. As SCZ patients were on average older than controls (effect size [e.s.] = 0.48 standard deviation (SD) units, p = 0.014), all statistical comparisons were performed using logistic regression with age and sex as covariates (see Materials and methods for details).

### Sleep stage macro-architecture is largely unaltered in SCZ

Based on manual sleep staging in 30 s epochs (see Materials and methods), the duration and proportion of sleep stages did not differ between SCZ and CTR groups, with the exception of the proportion of N2 sleep relative to total sleep time (TST), which was smaller in SCZ (e.s. = –0.38 SD, p = 0.048). For the primary analytic sample, we removed four subjects (due to persistent line noise artifacts, see Materials and methods for details); in this final $N$ = 126 sample, there were no significant case/control differences in stage duration (p = 0.14 for N2 percentage).

Patients spent longer time in bed (TIB) before sleep onset (sleep latency, 50 versus 24 min, e.s. = 0.95 SD, p = 0.0003), with concomitant differences in TIB (e.s. = 1.47 SD, p = $2\times10^{-7}$) and sleep efficiency (e.s. = –0.8 SD, p = 0.0013). Wake after sleep onset (WASO) time was not significantly different between groups (p = 0.13), suggesting that reduced sleep efficiency in SCZ was attributed to later sleep onset rather than more fragmented sleep. As a confirmation, group differences in sleep efficiency were not significant after controlling for sleep latency (p = 0.1).

Congruent with the sleep EEG night, data from a sleep habit questionnaire, a 2-week sleep journal, and information on sleep and wake times the night before the EEG recording all indicated that SCZ patients had earlier bedtimes (all p < $10^{-10}$) and longer TIB (all p < $10^{-8}$), suggesting that the EEG night followed a typical sleep/wake schedule for SCZ patients.

### N2 sigma power is decreased in SCZ

Sleep EEG analyses were based on 57 channels, resampled to 200 Hz and re-referenced to the linked mastoids. We extracted all N2 epochs and applied automated artifact detection/correction; all sleep EEG analyses used our open-source Luna package (see Materials and methods). With the exclusion of four patients due to persistent line noise artifacts, the final sleep EEG dataset comprised of 68 SCZ and 58 CTR, with an average of 353 ± 143 and 374 ± 89 N2 epochs, respectively (no group difference, p = 0.16).

Replicating prior studies (*Ferrarelli et al., 2007*; *Ferrarelli et al., 2010*; *Manoach et al., 2010*; *Manoach et al., 2014*), sigma-band (11–15 Hz) spectral power, a common proxy for spindle activity, was significantly reduced in SCZ (for power differences in classical frequency bands, see *Figure 1— figure supplement 1*). *Figure 1* (top row) illustrates spectral power from 0.5 to 20 Hz averaged over Fz, Cz, and Pz channels, with shaded regions indicating significant group differences (p < 0.01, unadjusted for multiple comparisons). Across all channels, the largest sigma-band deficit (–0.88 SD) was observed for FC1 at 13.25 Hz. Adjusted for multiple comparisons (see Materials and methods), 34

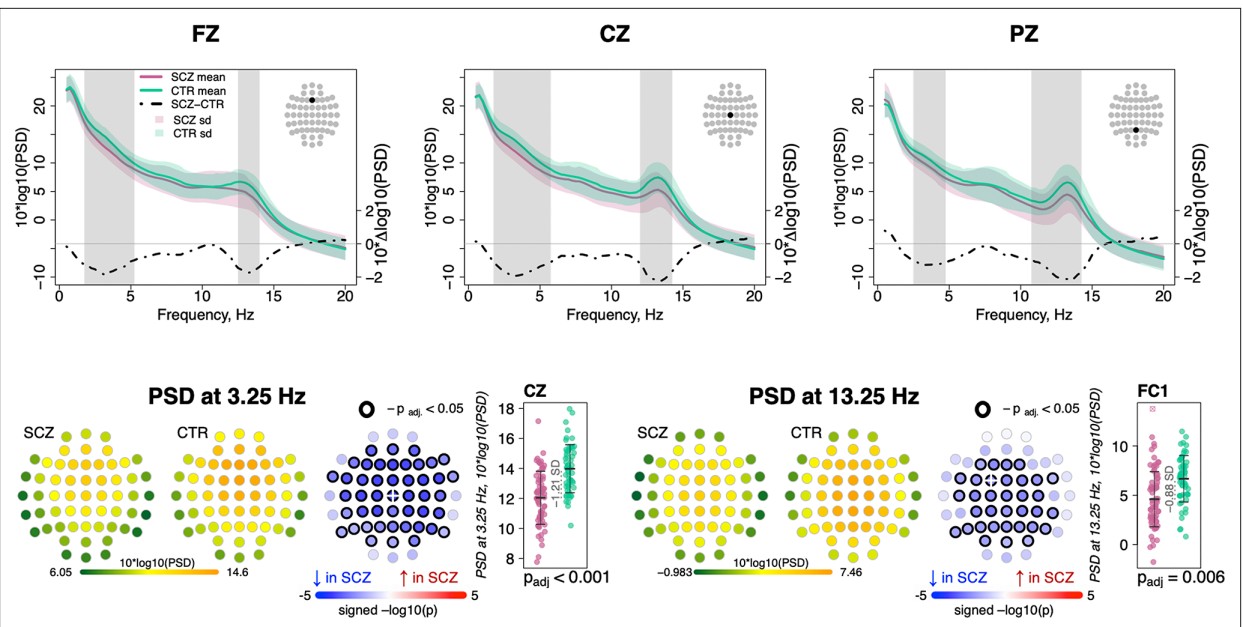

**Figure 1.** Decreased power in 2–6 Hz and spindle frequencies in schizophrenia (SCZ) during N2 sleep. The top row illustrates spectral power for frequencies from 0.5 to 20 Hz at Fz, Cz, and Pz channels averaged for SCZ (purple) and CTR (green) groups. Dotted black line illustrates the difference (SCZ-CTR) in averaged spectral power (10×Δlog10(PSD)) and frequency ranges marked with vertical gray bars illustrate frequency bins that showed significant group differences (p < 0.01, unadjusted). The plots in the bottom row illustrate power spectral density (PSD) averaged within groups and spatial distribution of group differences across channels. Channels with p-values adjusted for multiple comparisons ($p_{adj}$) < 0.05 are encircled with black line. The color of each channel corresponds to the signed −log10 adjusted p-value indicating the direction of group differences (blue corresponds to reduction, and red corresponds to increase in SCZ). Two scatterplots show individual spectral power for SCZ and CTR groups for channels with the largest effect sizes for the two indicated frequencies (marked with a white cross on the topoplots).

The online version of this article includes the following figure supplement(s) for figure 1:

**Figure supplement 1.** Topographical distribution of group differences in spectral power with respect to classic frequency bands.

channels showed deficits ($p_{adj}$ < 0.05) at 13.25 Hz (**Figure 1**, bottom right). Power in delta and lower theta frequencies (~2–6 Hz) was also reduced in SCZ (**Figure 1**, top row) with a maximum effect size at 3.25 Hz for Cz (e.s. = −1.21 SD) and 46 channels showing significantly ($p_{adj}$ < 0.05) lower power in SCZ (**Figure 1**, bottom left).

## Spindle density is reduced in SCZ, accompanied by altered spindle morphology

We detected discrete spindle events using our previously reported wavelet-based algorithm (**Purcell et al., 2017**), targeting SS and FS separately by setting wavelet center frequencies to 11 and 15 Hz, respectively. Spindle densities showed the expected topographies, being maximal at either central/ parietal (FS) or frontal (SS) channels (**Figure 2**). FS density was globally decreased in SCZ, significant at $p_{adj}$ < 0.05 for 53 of 57 channels, with the largest deficit (32% reduction and e.s. of −1.27 SD) at FC2 (nominal p = 4×10⁻⁶). At this channel, average FS density in the CTR group was 2.7±0.7 spindles per minute, compared to 1.9±0.8 in the SCZ group. SS also exhibited reduced densities, albeit mostly restricted to posterior channels (max e.s. −0.79 SD at P7, reduction = 28%, nominal p = 5×10⁻⁵). Taken together, these analyses confirmed reductions in both sigma power and spindle density with high statistical confidence, providing a clear and independent replication of prior reports.

For each spindle, we further estimated the amplitude (maximal peak-to-peak voltage), duration, integrated spindle activity (ISA, a normalized measure that reflects both duration and amplitude), frequency, and chirp (intra-spindle change in frequency, indexing the typical deceleration of a spindle oscillation over its course) (**Figure 2, Supplementary file 1**). See Materials and methods for details.

For SS, we observed an extensive SCZ-associated reduction in amplitude (50 channels with $p_{adj}$ < 0.05, max e.s. −0.82 SD with 18% reduction at CP3) and ISA (33 channels with $p_{adj}$ < 0.05, max e.s. −0.96 SD at CPZ). Reductions for FS were topographically more limited for amplitude (23 central/

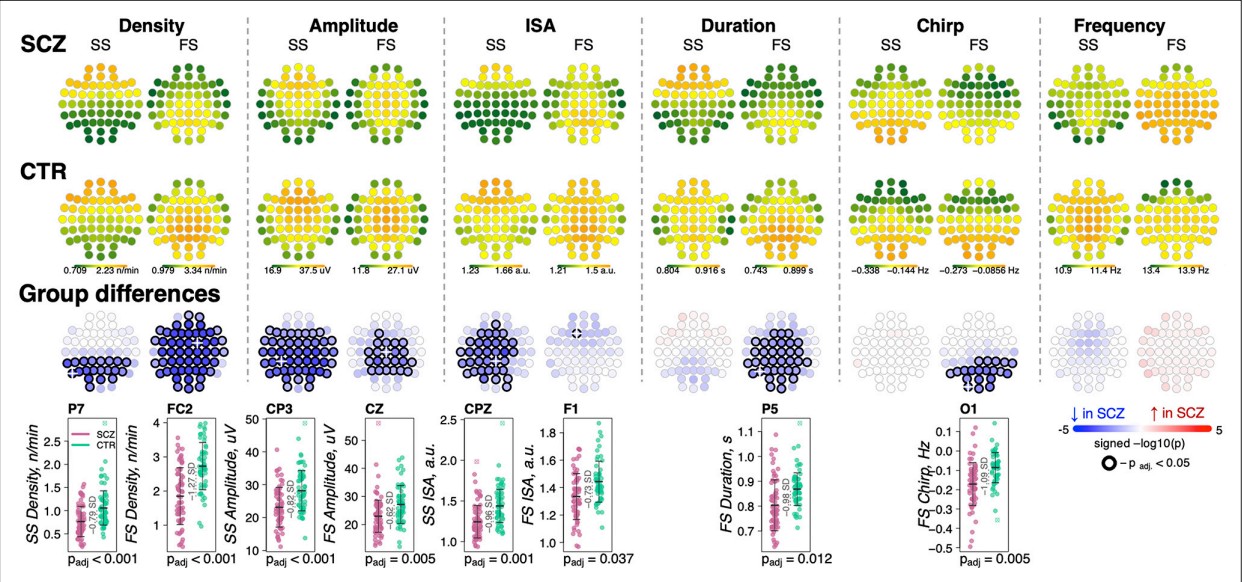

**Figure 2.** Reduced density of slow (SS) and fast spindles (FS) during N2 sleep in schizophrenia (SCZ) patients observed together with alterations in spindle morphology. The top two rows show topographical distribution in density, amplitude, integrated spindle activity (ISA), duration, chirp, and frequency of SS and FS averaged across SCZ (first row) and CTR (second row) groups. The third row of topographical plots illustrates group differences in these metrics. Each circle represents an EEG channel. EEG channels with adjusted p-values (multiple comparisons) < 0.05 are encircled in black and the color of each channel corresponds to the signed −log10 adjusted p-value indicating the direction of group differences (blue corresponds to reduction, and red corresponds to increase in SCZ). The bottom row illustrates distributions of spindle parameters in SCZ and CTR groups in the channel with the largest effect size of group differences (marked with a white cross on the topoplot).

The online version of this article includes the following figure supplement(s) for figure 2:

**Figure supplement 1.** Spindle and slow oscillation dynamics across the night.

**Figure supplement 2.** Associations between slow spindle density/amplitude and positive and negative syndrome scale (PANSS) scores.

**Figure supplement 3.** Spindles with different target frequencies.

parietal channels with $p_{adj} < 0.05$, max e.s. −0.62 SD with 15% reduction at Cz) and ISA (single channel, F1, $p_{adj} = 0.037$, e.s. = −0.73 SD).

FS (but not SS) were shorter in duration for SCZ in all but prefrontal and temporal channels (36 channels with $p_{adj} < 0.05$, max e.s. −0.98 SD at P5). In addition, FS chirp was more negative in SCZ (20 posterior channels with $p_{adj} < 0.05$, max e.s. −1.09 SD at O1), indicating more prominent deceleration in SCZ, that is, stronger chirp. Both SS and FS showed comparable SCZ and CTR distributions of observed average spindle frequencies, however.

Densities of SS and FS were not significantly correlated with each other ($abs(r) < 0.15$ and $p > 0.05$ for SS and FS at Cz or Fz, as well as SS at Fz compared to FS at Cz). With respect to density reduction in SCZ, SS and FS effects were also statistically independent. For example, group differences in SS density at P7 (largest e.s. of the group differences, $p = 5 \times 10^{-5}$) were still significant ($p = 6 \times 10^{-4}$) when FS density (at P7 and FC2 – largest e.s. of differences in FS density) was added as a covariate in a joint model. The same was true for FS density at FC2 (original $p = 4 \times 10^{-6}$ and after controlling for SS density, $p = 4 \times 10^{-5}$).

Below (see section *Dimension reduction to summarize sleep EEG alterations*), we explore in more detail the correlational structure of sleep EEG metrics, quantifying the underlying, independent components of variation across spectral, spindle, and other metrics, accounting for dependencies across scalp topography and neighboring frequencies.

We confirmed that observed spindle density alterations extended to entire NREM sleep (N2 and N3 combined, data not shown) but expressed a sleep cycle-dependent effect for both SS and FS with more extensive group differences in the later sleep (cycle 3 compared to cycle 1, *Figure 2—figure supplement 1*).

## Sensitivity analyses to address potential medication effects

Since 67 of 68 patients in the analytic sample were taking antipsychotic medication (*Supplementary file 2*), we converted antipsychotic doses to chlorpromazine equivalents (*Langan et al., 2012*), to determine whether group differences in spindle characteristics were likely to reflect medication effects (*Supplementary file 2*). There were no significant (all unadjusted p > 0.01) associations between total antipsychotic dose and any of the spindle parameters altered in SCZ, with the exception of FS chirp. However, this latter effect, seen only at two channels, had a direction of effect opposite to group differences observed in SCZ (FC6, *t*-value = −2.83, p = 0.006 and TP8, *t*-value = 2.8, p = 0.007), meaning that the original FS chirp SCZ association did not simply reflect antipsychotic dose.

Further, although multiple spindle parameters correlated with clozapine usage (*N* = 12 cases, *Supplementary file 2*), all original SCZ-CTR differences remained significant after patients using clozapine were removed from the analyses (*Supplementary file 3*). With respect to adjunctive medication such as sedatives and tranquilizers, mood stabilizers and antiepileptics, and anticholinergics (*Supplementary file 3*, top 8 rows), group differences in spindle metrics persisted after controlling for each class, except for FS ISA.

Taken together, these results strongly suggest that the observed group differences cannot be directly explained by medication effects, consistent with prior studies showing spindle deficits in unmedicated patients and first-degree relatives (*Manoach et al., 2014*).

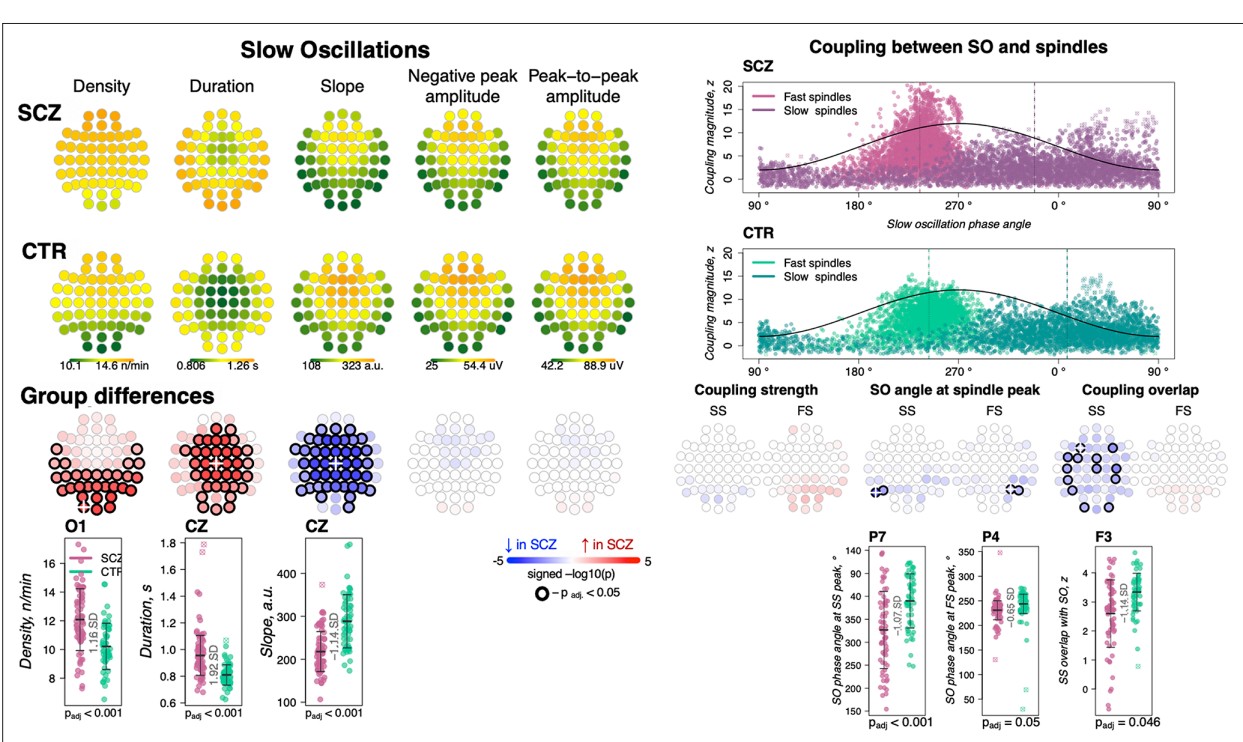

**Figure 3.** Altered slow oscillations (SOs) and their coupling with spindles in schizophrenia (SCZ). Left: Rows represent the topographical distribution of different SO parameters averaged across SCZ (first row) and CTR (second row) groups. Right: Coupling between SO and spindles in SCZ and CTR groups. Each point represents coupling magnitude and average SO phase of a spindle peak at a given channel in each participant. Mean SO angle of coupling for slow and fast spindles is shown as a vertical line of the corresponding color. The black line illustrates SO waveform at corresponding phase angles. Topoplots in the third row illustrate significance and direction of group differences across electroencephalogram (EEG) channels in SO parameters and its coupling with spindles. EEG channels with significant group differences ($p_{adj} < 0.05$) are encircled in black and the color of each channel represent the signed −log10 adjusted p-value indicating decrease (shades of blue) or increase (shades of red) in the tested EEG parameter in SCZ compared to CTR. Individual data at channels with the largest effect sizes of group differences are provided in the bottom row of scatter plots for the altered EEG parameters in SCZ (the exemplary channel is marked with a white cross on the topoplot above).

The online version of this article includes the following figure supplement(s) for figure 3:

**Figure supplement 1.** Topographical distribution of coupling characteristics averaged across schizophrenia (SCZ) and CTR groups.

## SO abnormalities in SCZ

We detected SOs in N2 sleep using a previously described heuristic, based on relative/adaptive amplitude thresholds (see Materials and methods). Patients had an increased density of SOs across posterior channels (30 channels with $p_{adj} < 0.05$, max e.s. = 1.16 SD at O1, 18% increase), longer SO duration/wavelength (36 channels with $p_{adj} < 0.05$, max e.s. 1.92 SD at Cz, 18% increase) and flattened SO slope (44 channels with $p_{adj} < 0.05$, max e.s. −1.14 SD at Cz, 24% decrease) compared to controls (*Figure 3*, left panel). No differences between groups were observed in negative peak or peak-to-peak amplitude.

SO parameters were not associated with total dose of antipsychotics but were affected by adjunctive medications: namely, increased SO duration and decreased slope were significantly associated with use of sedatives and tranquilizers; reduced SO slope was also linked to mood stabilizing and antiepileptic medication (uncorrected p < 0.01, *Supplementary file 2*). However, all SCZ-CTR group differences persisted in sensitivity analyses controlling for medication dosage (*Supplementary file 3*).

Of note, group differences in SO characteristics depended on the use of absolute versus relative amplitude thresholds for detecting SOs, as we have noted in other contexts (*Djonlagic et al., 2021*). We adopted relative amplitude thresholds for our primary analyses, as it maximized the magnitude of spindle/SO coupling observed across the whole sample. If using an absolute threshold, differences in SO density, duration, and slope were attenuated, whereas SO negative peak amplitude achieved significant group differences.

In addition, as SOs are most characteristic of N3 sleep, we performed secondary analysis that combined N2 and N3 sleep periods. While alterations in duration and slope of SOs remained significantly different between the groups, there were no differences in SO density applying either absolute or adaptive amplitude threshold for SO detection. A cycle specific analysis of N2 using adaptive threshold revealed that SO density alterations were more pronounced later in the night (*Figure 2—figure supplement 1*).

## Preserved coupling strength but altered overlap between SS and SO

We quantified SO/spindle coupling in three ways, using empirically derived surrogate distributions to control for chance coupling (see Materials and methods): the rate of gross overlap between spindles and SOs, the magnitude of coupling based on the non-uniformity of the distribution of SO phase at spindle peaks, and the mean phase angle derived from that distribution.

As expected, we observed a marked, non-random temporal coupling between both FS and SS and SO phase (*Figure 3*, top right). Consistent with previous reports (*Djonlagic et al., 2021*) whereas FS peaks (points of maximal peak-to-peak amplitude) were enriched on the rising slope of the SO and near the SO positive peak (mean phase = 240.8 degrees in CTR), SS tended to peak later (mean phase = 13.8 degrees in CTR) (*Figure 3*, top right). In CTRs, 97% of subjects had significant SO phase coupling for FS at Cz (empirical p-value < 0.05 obtained by randomly shuffling spindle peak timing with respect to SO phase to generate the null distribution) and 76% had significant SS coupling at Fz. The SCZ group displayed broadly comparable proportions (94% and 68% of cases showing evidence of non-uniform coupling, for FS at Cz and SS at Fz, respectively).

We did not observe any significant group differences in the average magnitude of SO phase/spindle coupling based on the surrogate distribution normalized metric (*Figure 3*, right panel; see *Figure 3—figure supplement 1* for the topographical distribution of coupling characteristics averaged across SCZ and CTR groups). With respect to SO phase, both SS and FS tended to occur earlier in SCZ, albeit this effect was only observed at a few parietal channels therefore precluding any strong conclusions (P5 and P7 for SS, max e.s. = −1.07 SD and P4, P6 for FS, max e.s. = −0.65 SD). The most pronounced coupling differences were in gross overlap of SS (the number of spindles showing any overlap with an SO, controlling for spindle and SO density), which was decreased in SCZ (13 channels with $p_{adj} < 0.05$, max e.s. −1.14 SD at F3).

None of the coupling metrics were associated with total dose of antipsychotics but there was increased SS-SO overlap in patients taking aripiprazole or clozapine; an opposite effect was observed for mood stabilizing and antiepileptic medication (*Supplementary file 2*).

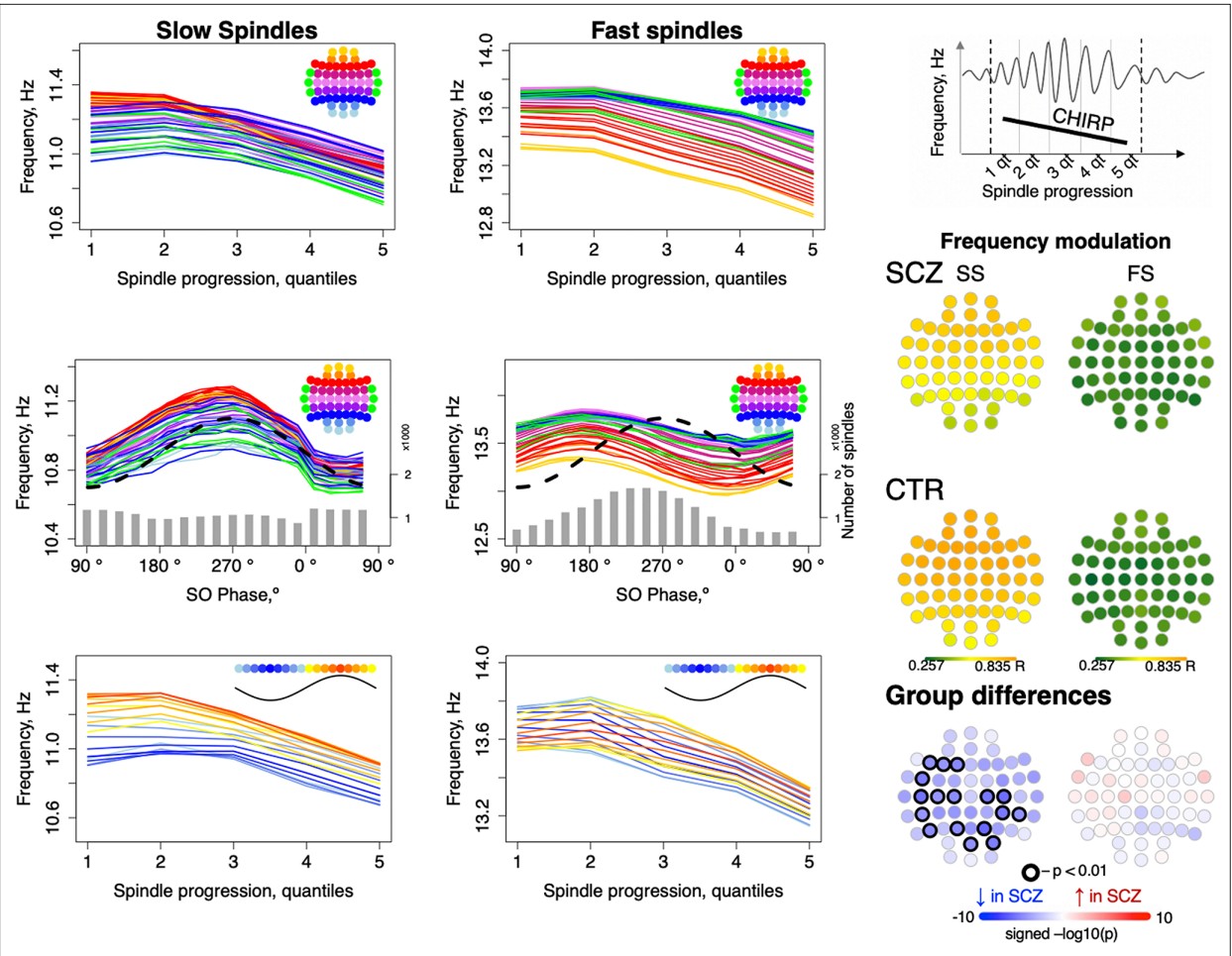

**Figure 4.** Intra-spindle frequency deceleration (chirp) is associated with slow oscillation (SO) phase differently for slow and fast spindles and such phase-frequency modulation is attenuated in schizophrenia (SCZ). Top: frequency changes across slow and fast spindle as spindles progress in five quintiles across all channels (averaged across all participants). Middle: intra-spindle frequency dependency on SO phase across all channels (lines) and total number of spindles detected in all channels and individuals at a given SO phase bin (gray bars). Bottom: spindle frequency as a function of both spindle progression (y axis) and SO phase (color of the curve; each curve represents frequency of spindles averaged across all channels for each 20-degree phase bin; blue curves illustrate frequency of spindles occurring close to the negative peak and red curves illustrate spindles next to the positive peak of SOs). The topoplots on the right illustrate topographical distribution of phase/frequency modulation averaged within SCZ and CTR groups separately for slow and fast spindles. Phase/frequency was estimated as linear-circular correlation between instantaneous frequency of individual spindle and phase of co-occurring SO. Topoplots in bottom right illustrate group differences in phase/frequency modulation. Channels with uncorrected p-values < 0.01 are encircled in black, and the color of each channel represents the signed −log10 p-value indicating decreased (shades of blue) or increased (shades of red) coupling in the SCZ group.

The online version of this article includes the following figure supplement(s) for figure 4:

**Figure supplement 1.** Instantaneous frequency and its coupling with slow oscillation (SO) phase.

### Intra-spindle frequency modulation: chirp/deceleration and SO phase

As noted, we observed a greater deceleration of FS (chirp) in SCZ. This effect was statistically independent of the reduction in FS density: for example, in a joint model, FS chirp and density at Pz were both independent predictors of SCZ (FS density $t$-score = –3.77, $p$ = 0.0002 and chirp $t$-score = –3.17, $p$ = 0.0015).

Given that spindles are temporally coupled with SO phase, as well as the fact that we observed SCZ/CTR differences for both FS chirp and preferential SO phase at spindle peak, we asked whether spindle deceleration was dependent on SO phase. We summarized spindle instantaneous frequency (estimated using a filter-Hilbert approach, see Materials and methods) as a function of spindle progression (quantiles of spindle duration) and also SO phase (in eighteen 20-degree bins).

Considering first only spindle progression, we observed the characteristic negative chirp (deceleration during the later portion of the spindle), for both FS and SS across all channels (*Figure 4*, top row). We further observed that spindle frequency was modulated by SO phase, consistently across channels but differently for FS and SS, likely reflecting the different temporal SO phase coupling of FS and SS (*Figure 4*, middle row): for SS, the highest instantaneous frequencies coincided with the SO positive peak, but earlier for FS (*Figure 3*, top right).

Spindle progression and SO phase were related, reflecting SO/spindle coupling. We therefore additionally summarized spindle instantaneous frequency as a joint function of both spindle progression and SO phase, that is, 5 spindle progression quantiles x 18 SO-phase bins = 90 combinations averaged over all channels (*Figure 4*, lower row, *Figure 4—figure supplement 1*). This analysis suggested that spindle progression and SO phase have independent, largely additive effects on spindle frequency, as each line (SO phase bin) shows broadly similar chirp. Likewise, conditional on spindle progression, the SO modulation effect is still evident by the vertical offsets of the different lines (SO phase bins).

More formally, modeling SS instantaneous frequency as a cubic function of spindle progression and/or SO phase (CTRs only) showed an associated adjusted $R^2$ = 0.48 and 0.44, respectively, and $R^2$ = 0.95 if combined. For FS, SO phase explained relatively less of the variance in spindle frequency ($R^2$ = 0.19 versus $R^2$ = 0.72 for spindle progression; combined, $R^2$ = 0.94). That is, SO phase has a modulating effect on spindle frequency, independent of the typical deceleration/chirp, and this modulation was stronger for SS than FS. To distinguish this from temporal spindle/SO coupling, we refer to it as 'phase/frequency modulation', whereby the frequency of spindle oscillations depends on SO phase in

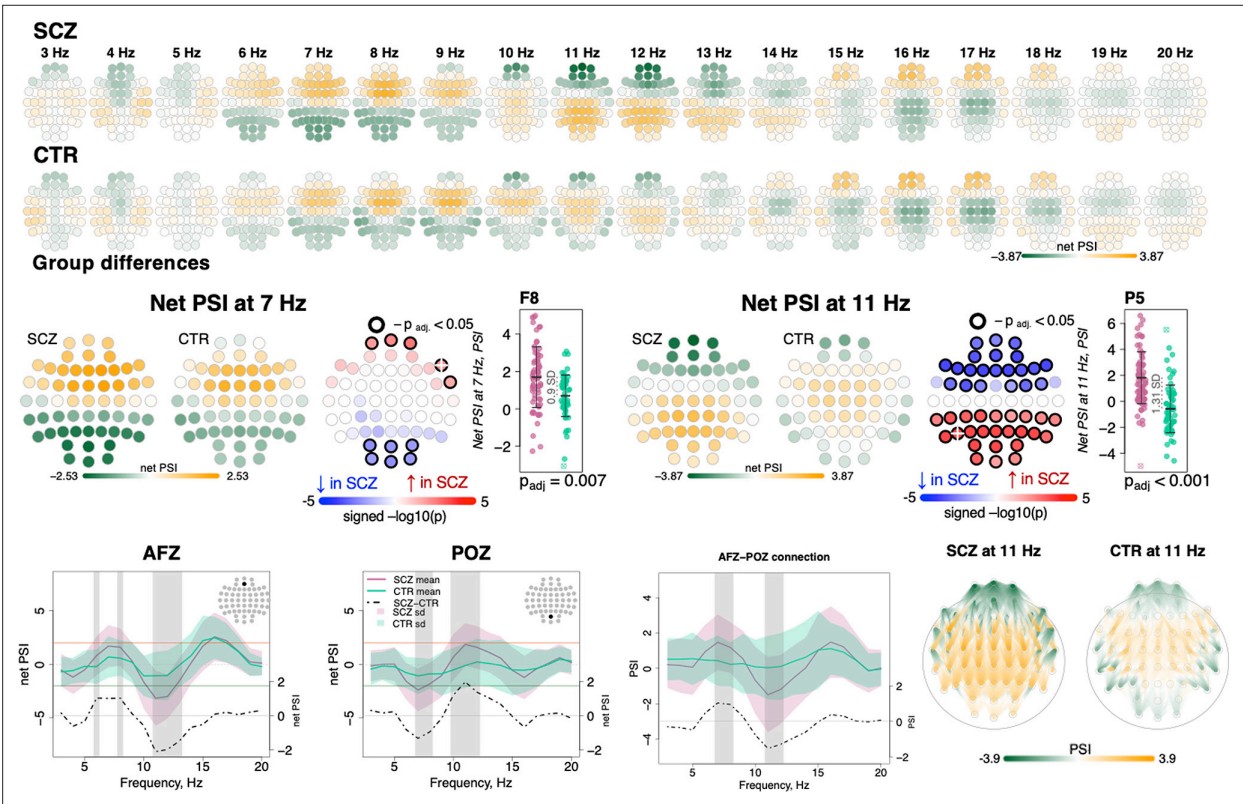

**Figure 5.** Directionality of information flow and hyperconnectivity in schizophrenia (SCZ). The top two rows illustrate net phase slope index (PSI) values across channels for each center frequency from 3 to 20 Hz averaged in SCZ and CTR groups. The topoplots in the middle row provide illustration of topographical distribution of net PSI values and group differences at 7 and 11 Hz (frequency bins where largest effect sizes of group differences in net PSI were observed). Channels with significant group differences ($p_{adj}$ < 0.05) are encircled with black line and the color of channels represents signed –log10 p-value with shades of blue indicating decrease in SCZ and shades of red indicating increase in SCZ. Three plots from the left in the bottom row illustrate averaged net PSI and group differences (shadow regions where unadjusted p-values < 0.01) for AFz and POz and pairwise PSI values between AFz and POz. The topoplots on the right illustrate PSI connectivity at 11 Hz averaged for SCZ and CTR groups between channel pairs with uncorrected p-values < 0.001.

a temporally fine-grained manner, which could be mediated via differential hyperpolarization associated with global up- and down-states.

To quantify SO/spindle phase/frequency modulation for each individual/channel, we calculated linear-circular correlations between binned instantaneous spindle frequency and SO phase (see Materials and methods) and observed reductions in SO phase/SS frequency modulation in SCZ (*Figure 4*, right panel).

## Functional hyperconnectivity in SCZ during N2 sleep

To index functional connectivity we computed the phase slope index (PSI) (*Nolte et al., 2008*) between all channel pairs for frequencies from 0.5 to 22.5 Hz in 5 Hz windows (resulting from center frequencies from 3 to 20 Hz shifted by 1 Hz increments). For each channel, we summed the normalized PSI of all pairs involving that channel: higher absolute values for this net PSI metric indicate that on average the channel tended to be functionally connected to other channels in a way that was either leading (positive values) or lagging (negative values) other channels with respect to coherent phase differences.

Comparing SCZ and CTR, directional net connectivity followed a qualitatively similar pattern in both groups, whereby frontocentral channels showed positive and posterior channels showed negative net PSIs at 6–9 and 16–17 Hz (center frequencies); in contrast, for 11–13 Hz, centroparietal channels had positive and frontal channels had negative net PSIs (*Figure 5*, two top rows). Such connectivity patterns were observed at the channel level in net PSI as well as in pairwise PSI (*Figure 5*, bottom row displays net PSI for two exemplary channels – AFz, POz – and pairwise PSI between them).

Despite broadly similar patterns, significant group differences in net PSI were evident for 6–8 and 10–13 Hz (shaded areas at the channel level plots in *Figure 5*, bottom row, and topoplots *Figure 5*, middle row). Within the 6–8 Hz interval, five frontal channels had higher net PSI in SCZ, whereas six occipital channels had more negative values ($p_{adj}$ < 0.05, *Figure 5*, middle left, largest e.s. = 0.9 SD for F8 at 7 Hz). In both cases, this corresponded to greater connectivity in SCZ (i.e. larger absolute PSIs). More extensive group differences were observed at 11 Hz (*Figure 5*, middle right) with 21 anterior channels showing more negative and 24 posterior channels with more positive net PSIs (largest e.s. = 1.31 SD at P5). A similar pattern of differences was observed if pairwise PSI values were tested; again, both effects were consistent with stronger connectivity (higher absolute net PSI values) in SCZ.

We also tested group differences in sigma-band magnitude squared coherence – the most studied aspect of connectivity in sleep in SCZ so far (*Wamsley et al., 2012*; *Sun et al., 2021*; *Markovic et al., 2020*) – but found no evidence of significant alterations. Out of 1596 channel pairs, only 53 expressed group differences with uncorrected p-value < 0.05, which is lower than expected by chance (1596×0.05~80 significant pairs). We also specifically tested connections between the channels previously reported to have altered coherence in SCZ. For example, decreased coherence between C3 and O2 was reported in both *Wamsley et al., 2012*; *Sun et al., 2021*, but it was not different in our sample (e.s. –0.06 SD, nominal p-value = 0.98). Likewise, we observed no reduction in coherence between C3 and O1 (e.s. –0.05 SD, nominal p-value = 0.95) that was shown in both *Sun et al., 2021*; *Markovic et al., 2020*.

## Dimension reduction to summarize sleep EEG alterations

To summarize the many EEG metrics (across domains but also channels and frequencies), we applied singular value decomposition (SVD), as previously applied for the sleep EEG where it was called principal spectral component (PSC) analysis (*Djonlagic et al., 2021*). PSC analysis typically yields a small number of orthogonal components (linear combinations of the original features) which can be used as new summary score metrics (see Materials and methods). For example, applied to the 4503 spectral power metrics (57 channels by 79 frequency bins between 0.5 and 20 Hz), 11 orthogonal components explained over 90% of the total variance (*Figure 6*). When based on power spectra from N2 sleep, many components (especially the larger ones) have relatively straightforward interpretations in terms of patterns of spectral or spatial profiles, or in terms of related sleep EEG metrics such as spindle parameters. Here, the first component indexed variation in total power – as evident from inspection of the component loadings (*Figure 6*) and strong correlation between the component scores and total power (0.5–20 Hz) at Cz yielding correlation coefficient of 0.95 and p-value = $1 \times 10^{-62}$ (note that the sign of PSCs is not defined).

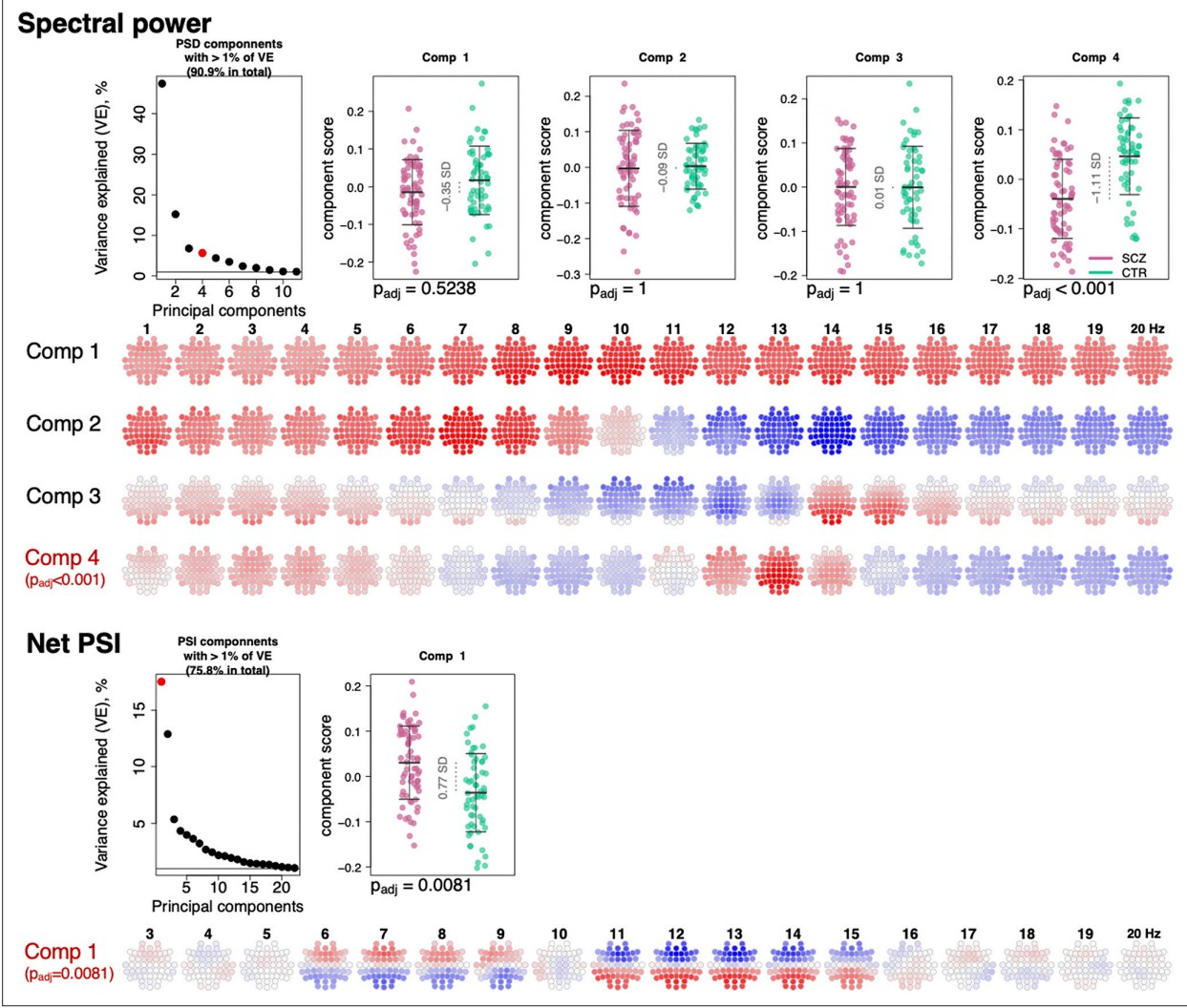

**Figure 6.** Power spectral density (PSD) and net phase slope index (PSI) components recapitulate alterations in spectral power and net PSI. The top left plot illustrates variance explained by the first 11 PSD components (thresholded at 1% variance explained). Red color highlights the component that expressed significant association with schizophrenia (SCZ) ($p_{adj} < 0.05$ Bonferroni adjusted for 11 comparisons). Scatterplots to the right illustrate each component's score in SCZ and CTR groups. Four rows of topoplots on the bottom represent each component's loadings across channels and frequencies. The first component of PSI is presented in a similar fashion.

The second component, PSC2, was independent of total power (p-value = 0.87 for Cz in CTR group), reflecting the orthogonality of components, but was instead strongly associated ($r = –0.78$ and p-value = $4 \times 10^{-27}$) with the ~$1/f$ spectral slope, as estimated by a linear regression of log power on log frequency (0.5–20 Hz). PSC3 captured individual differences in spindle amplitude (*Figure 6*) correlating negatively with SS amplitude at Cz ($r = –0.31$, p-value = 0.0004) but positively with FS amplitude ($r = 0.32$, p-value = 0.0003) at Cz. Importantly, unlike sigma-band power, this component effectively controls for differences in total power and spectral slope.

All three components – that collectively accounted for 69% of the total variation in power – had similar distributions in SCZ and CTR. However, component 4, which accounted for 5.8% of the total variance, was significantly associated with SCZ ($p_{adj} = 1 \times 10^{-4}$ [Bonferroni corrected for 11 components], e.s. = 1.1 SD). This component captured coordinated SCZ-associated reductions in power at 2–5 and 12–14 Hz (*Figure 6*, topoplots corresponding to component 4 loadings across channels and frequencies), echoing our previous univariate analyses pointing to decreased power in sigma and delta frequency ranges (*Figure 1*). Unlike component 3, component 4 was more strongly associated with spindle density than amplitude, and more so for FS (SS density $r = 0.3$, p-value = 0.0005 and FS density $r = 0.6$, p-value = $2 \times 10^{-13}$, SS amplitude p-value > 0.05, FS amplitude p-value = 0.04).

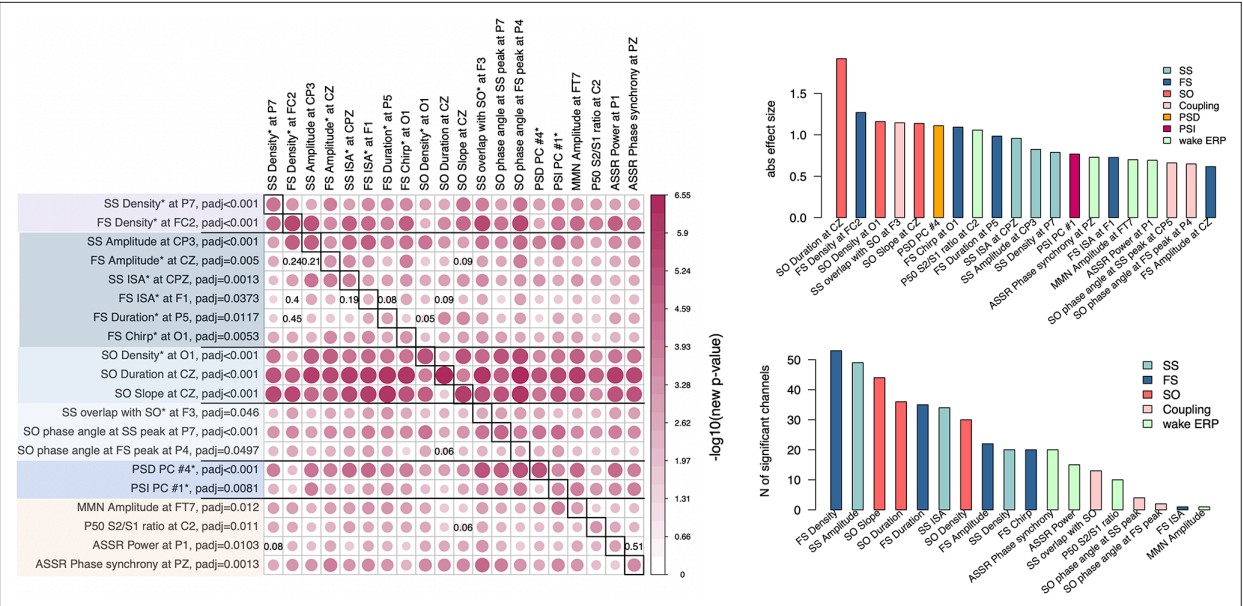

**Figure 7.** Joint model analysis across electroencephalogram (EEG) metrics that were altered in schizophrenia (SCZ). The matrix on the left illustrates p-values of the joint model analysis. A joint logistic regression model with formula logit(DISORDER) ~ EEG metric1 at CH1 + EEG metric2 at CH1 + EEG metric2 at CH2 + sex + age + error was fitted for each EEG metric, where CH1 is the channel with the largest effect size for EEG metric1 and CH2 is the channel with the largest effect size for EEG metric2. EEG metric1 is specified by rows, EEG metric2 by columns. The relative size and color in each cell indicate −log10 unadjusted p-value in EEG metric1 at the CH1 after EEG metric2 at CH1 and CH2 was added as a covariate to the logistic regression. p-Values of EEG metric1 where group differences were not significant (p > 0.05) after covariates were added are provided in corresponding cells. Diagonal cells contain the original (i.e. not adjusted by metric2) p-values for metric1. Two bar plots on the right illustrate absolute effects sizes (top) or number of significant ($p_{adj} < 0.05$) channels among the tested metrics (the bar plot on the bottom right does not include power spectral density [PSD] and phase slope index [PSI] because these were based on principal spectral components (PSCs) and the number of channels could not be directly estimated).

Applying the same approach to the net PSI-based functional connectivity metrics, we reduced 1026 variables (57 channels, 18 frequency bins) to 22 independent components that each accounted for at least 1% of the total variance (76% in total). The first component explained 17.6% of the variance and was significantly different between SCZ and CTR ($p_{adj}$ = 0.0081 [Bonferroni corrected for 22 comparisons], e.s. = 0.76 SD). This component captured spatially higher connectivity at 6–9 and 11–15 Hz, with higher values in SCZ indicating hyperconnectivity, recapitulating our previous results (*Figure 5*).

Thus, PSCs were able to capture SCZ-related alterations in spectral power and net PSI while reducing the dimensionality of the data. As well as providing a framework for multiple testing correction, these analyses can indicate which changes are independent versus linked: for example, here the reductions in delta and sigma-band power map onto the same component. As described below, we used selected components to determine how different domains of sleep microarchitecture were related to each other and to the wake ERP; we also used components to build joint models to predict disease status.

## Joint model analysis of sleep EEG alterations suggests their non-redundancy

Given that we observed SCZ-related alterations across multiple facets of the sleep EEG, we tested the extent to which they were independent of each other. Taking the most significant channel for each associated metric (e.g. SS density at P7, or FS density at C1) yielded 16 sleep EEG variables (*Figure 7*); for each of these we queried whether conditioning on the remaining 15 variables (sequentially) accounted for the original association with SCZ, defined as a nominal p > 0.05. With a few exceptions, most of the effects were independent of each other. Briefly, FS amplitude effects were accounted for by either FS density, SS amplitude, or SO slope; FS ISA and duration were accounted for by FS density or SO density; FS/SO phase angle effects were accounted for by SO duration. All other effects (12 out of 16 sleep EEG metrics) remained significant no matter which other metric was covaried for.

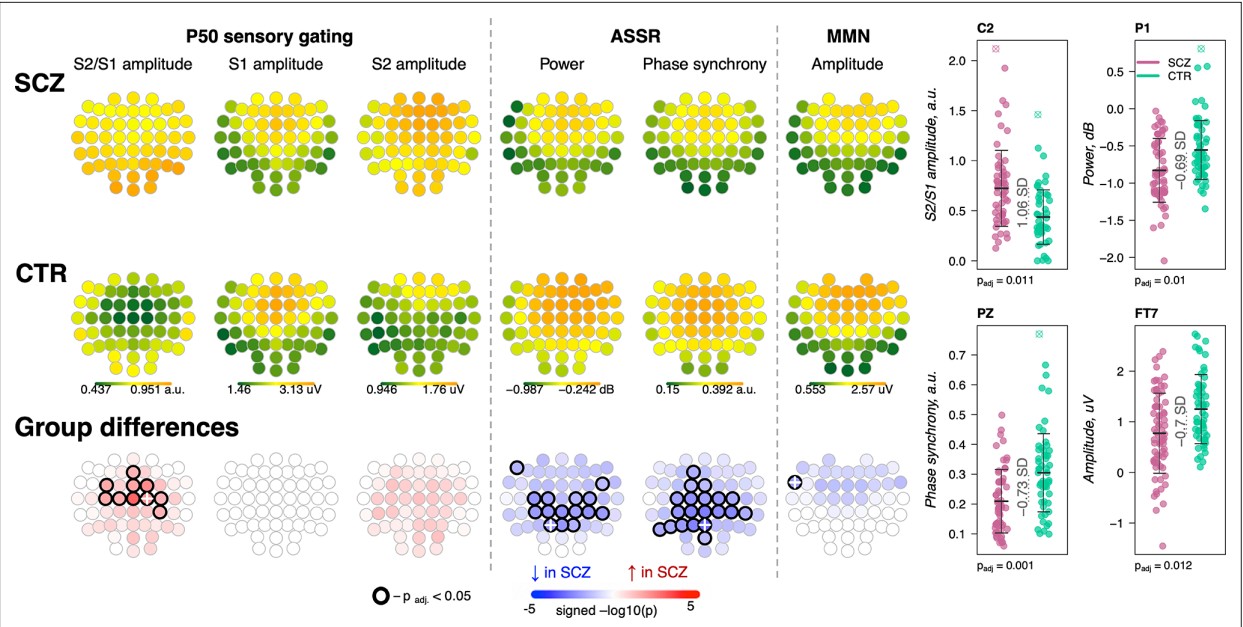

**Figure 8.** Awake auditory processing abnormalities in schizophrenia (SCZ): diminished mismatch negativity (MMN) amplitude, increased P50 ratio between S2 and S1 during sensory gating and reduced power and phase synchrony of auditory steady-state response (ASSR). The top two rows' topoplots represent the topographical distribution in P50 S2/S1 ratio, as well as the P50 amplitudes elicited by stimulus S1 and stimulus S2; ASSR power and ASSR phase synchrony; MMN amplitude averaged across patients with SCZ (the first row) and controls (the second row). The topoplots in the third row illustrate significance and direction of group differences across electroencephalogram (EEG) channels. Four scatterplots on the top right display distributions of each metric with significant group differences at an exemplary channel with the largest effect size (marked with a white cross on the topoplots illustrating group differences).

Pairwise analyses such as these do not provide a complete description of the multivariate structure of the data: nonetheless, these analyses clearly indicate that most metrics captured non-redundant features of the sleep EEG. Two important results emerge: (1) reductions in SS and FS density were independent of each other, (2) alterations in power, net PSI, SO, and most of the coupling parameters persisted over and above the spindle metrics (*Figure 7*). These multiple and (at least partially) independent associations may reflect distinct neurophysiology abnormalities underlying SCZ.

## Altered auditory processing in SCZ is mostly independent of sleep abnormalities

In the evening prior to the sleep study, participants performed three ERP paradigms, all of which have previously been reported to be altered in SCZ: sensory gating P50 (*Freedman et al., 2020*), mismatch negativity (MMN) (*Erickson et al., 2016*), and auditory steady-state response (ASSR) (*Thuné et al., 2016*).

Presented with identical consecutive auditory stimuli (S1 and S2 delivered 500 ms apart), typical sensory gating leads to suppression of the second P50 evoked response (S2) relative to the first (S1). SCZ patients showed reduced gating (higher S2/S1 ratios) in 10 frontocentral channels ($p_{adj} < 0.05$, max e.s. = 1.06 SD at C2) (*Figure 8*, left). Consistent with previous reports (*Atagun et al., 2020*; *Bramon et al., 2004*), this was primarily driven by elevated S2 amplitudes in SCZ.

The ASSR evaluates neural synchrony of the entrainment response elicited by 40 Hz auditory stimulation and is measured by evoked power and phase locking. Both measures were decreased in patients, predominantly in central and parietal channels (*Figure 8*). ASSR power differences were significant in 15 channels ($p_{adj} < 0.05$, max e.s. = –0.69 SD at C2), while phase locking differences were present in 20 channels ($p_{adj} < 0.05$, max e.s. = –0.73 SD at Pz).

MMN measures the response to infrequent deviant stimuli (auditory tones differing in a perceptual feature such as duration) compared to repeated standard stimuli, generally occurring ~150–250 ms after stimulus onset. MMN amplitudes were reduced in frontocentral channels in SCZ group (*Figure 8*), but only differences at FT7 survived multiple comparison correction ($p_{adj} = 0.012$, e.s. = –0.7 SD).

These results broadly replicate previous reports (*Erickson et al., 2016*; *Freedman et al., 2020*; *Thuné et al., 2016*) with patients showing reductions in sensory gating, MMN, and ASSR. Despite similar effect sizes for MMN and ASSR, the former showed an attenuated responses in terms of the number of channels surviving correction. Nevertheless, congruent with literature indicating central channels being the most affected in SCZ during auditory processing, all ERP metrics displayed nominally significant group differences at Cz: P50 ratio – e.s. = 0.92 SD, uncorrected p-value = 0.0002, MMN amplitude – e.s. = −0.49 SD, p-value = 0.008, ASSR power – e.s. = −0.56 SD, p-value = 0.008, and ASSR phase synchrony – e.s. = −0.62 SD, p-value = 0.002.

Notably, SCZ-associated wake ERP and sleep EEG metrics were independent of one another. Among 100 comparisons (4 wake by 25 sleep metrics at Cz), only one showed moderate dependency (defined here as an absolute correlation coefficient >0.2 and p < 0.01): P50 S2/S1 ratio and SO peak-to-peak amplitude ($r$ = −0.25, p = 0.01). This correlation between sensory gating and SO amplitude may be attributable to the shared effect of total power, as it was not significant (p > 0.05) after controlling for total power during N2 sleep.

## Sleep/wake EEG impairments and clinical variables

We used the positive and negative syndrome scale (PANSS) to characterize patient symptomatology based on a five-factor model (positive, negative, disorganized-concrete, excited, and depressed subscales) that has been shown to perform better than the traditional three PANSS subscales (*Wallwork et al., 2012*). Testing the 20 metrics associated with SCZ, reduced SS density was associated with disorganized-concrete scores (14 channels with $p_{adj}$ < 0.05, max e.s. $r$ = −0.43 p = 0.0003 at T8, *Figure 2—figure supplement 2*) and lower SS amplitude was associated with negative symptoms (six channels with $p_{adj}$ < 0.05, maximum effect size $r$ = −0.357 p = 0.003 at CP3, *Figure 2—figure supplement 2*). Other weaker but significant associations included: lower FS amplitude and MMN amplitude were associated with negative symptoms, and reduced SO density with a longer duration of disease (*Supplementary file 4*). These data suggest SS density and amplitude in particular indexed negative symptoms and disorganized thoughts in patients.

One recent report suggested that spindle deficit may disproportionally affect SCZ patients with auditory hallucinations (*Sun et al., 2021*). While the Auditory Hallucination Rating Scale was not available for our cohort, the PANSS, which assesses hallucinatory behavior (positive symptoms scale question 3, P3) (*Kay et al., 1987*), indicated that the majority of patients in our cohort (45 out of 70 patients for whom PANSS assessment was available) did not experience hallucinations. Spindle deficits in the subsample of hallucination-free patients were still significant and of comparable effect sizes (SS density at P7 e.s. = −0.52 SD, p = 0.01 versus full sample e.s. = −0.79 SD, p = 5.6×10$^{-5}$; FS density at FC2 e.s. = −1.36 SD, p = 2×10$^{-5}$ versus full sample e.s. = −1.27 SD, p = 4×10$^{-6}$). Association analysis indicated that PANSS P3 score in SCZ patients negatively correlated with SS density at P7 ($r$ = −0.32, p = 0.008), but there was no significant association between P3 score and FS density at FC2 ($r$ = 0.11, p = 0.37). Taken together, SS and FS density reduction was not restricted to patients with hallucinations in our cohort and once more only SS density exhibited a significant link with symptomatology.

## Independent replication of NREM sleep metric SCZ associations

Compiling whole-night sleep EEG recordings across three previously reported studies (*Mylonas et al., 2020a*; *Wamsley et al., 2012*; *Manoach et al., 2010*), we obtained a replication dataset of SCZ patients (*N* = 57) and controls (*N* = 59) (*Supplementary file 5*). All studies contained a common set of four channels (C3, C4, O1, and O2) although only *N* = 55 (26 SCZ, 29 CTR) had comparable hdEEG. Connectivity and SVD-based replication analyses were therefore restricted to that subset, whereas tests of spindles, SO, and coupling used the full four-channel replication sample. Associations between SCZ and several sleep parameters have been described previously for these studies individually, including spindle metrics, SO parameters, coupling, and spectral power (*Mylonas et al., 2020a*; *Wamsley et al., 2012*; *Manoach et al., 2010*; *Demanuele et al., 2017*). However, there has not been a comprehensive and joint analysis across datasets that is methodologically consistent with our sample (we refer to it as GRINS sample from this point on). For example, the prior studies only examined FS and did not measure spindle chirp or global connectivity (*Wamsley et al., 2012*).

Out of 16 sleep EEG metrics (*Figure 7*, left, rows with different shades of blue background), group differences in 11 sleep metrics were replicated in the independent sample (*Figure 7*, left, EEG

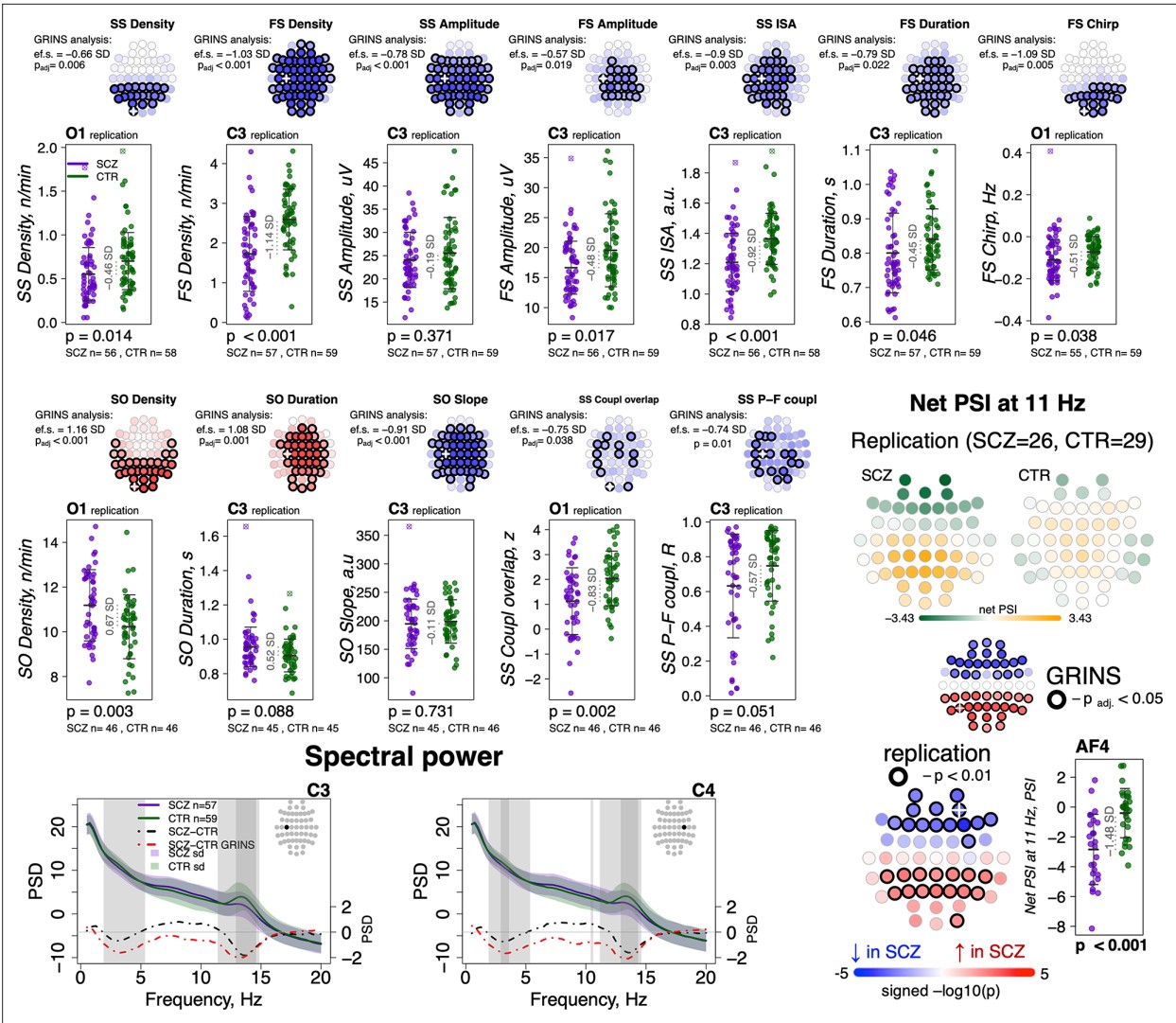

**Figure 9.** Most sleep microarchitecture alterations were replicated in an independent dataset. Top row – topoplots recapitulate the findings in spindle parameters observed during the original analysis in the GRINS cohort; channels marked with a white cross were tested in the replication cohort. p-Values and effect sizes from the between-group comparison of the GRINS sample is to the left of the topoplots. The scatterplots below illustrate the corresponding group differences in the replication datasets. Middle two rows – similar plots for slow oscillation (SO) and coupling parameters. Two graphs in the bottom row show averaged spectral power across groups (schizophrenia [SCZ] in purple, CTR in dark green, shadow of the same color represents standard deviation) for 0.5–20 Hz. The dotted lines illustrate difference in spectral power between the groups (SCZ-CTR) in the replication sample (black) and GRINS (dark red). Lighter shaded areas represent frequency bins where group differences were significant in GRINS (p < 0.01) and darker shaded areas mark frequency bins where group differences were observed in the replication sample (p < 0.05). Topoplots displayed in the bottom right corner illustrate averaged net phase slope index (PSI) values and group differences (uncorrected p < 0.01) in the replication sample with high-density electroencephalogram (EEG).

The online version of this article includes the following figure supplement(s) for figure 9:

**Figure supplement 1.** Net phase slope index (PSI) during N2 and REM in control subjects in GRINS and ESZ sample.

variables marked with a star). With respect to spindle metrics, most of primary findings from GRINS were replicated (nominal p < 0.05) in the independent dataset, the one exception being SS amplitude (GRINS C3: p < 0.001, e.s. = –0.78 SD, but replication C3: p = 0.37, e.s. = –0.19 SD) (**Figure 9**, two top rows, **Supplementary file 6**). We also replicated the association of increased SO density (GRINS O1 p < 0.001, e.s = 1.16 SD; replication O1, p = 0.003, e.s = 0.67 SD) and observed a consistent trend for SO duration (GRINS C3 p < 0.001, e.s = 1.08 SD; replication O1 p = 0.09, e.s = 0.52 SD). Differences in SO slope were not replicated, however. Alterations in coupling were replicated only for reduced SS/

SO overlap (GRINS O1 p = 0.002, e.s. = −0.75 SD; replication C3 p = 0.002, e.s. = −0.83 SD (*Figure 9*, middle two rows, *Supplementary file 6*).

With respect to spectral power, we replicated reductions in sigma power at C3 and C4, albeit with smaller effect sizes than seen in GRINS, whereas decreased delta power replicated only at C4 (*Figure 9*, bottom row left).

Replication analyses of net PSI connectivity yielded similar average patterns of directional connectivity as observed in GRINS (*Figure 9—figure supplement 1*). Of note, we also estimated mean net PSI topographies during REM sleep, which were also similar between GRINS and the replication cohort, but qualitatively different from NREM, suggesting that these connectivity profiles reflect functional connectivity patterns specific to NREM sleep, rather than arising obligatorily, for example, if reflecting only fixed aspects of brain morphology (*Figure 9—figure supplement 1*). We replicated our finding of elevated sigma-band (11 Hz center frequency) connectivity (uncorrected p < 0.01) at 11 Hz in SCZ (*Figure 9*, bottom row right), but the 7 Hz result in GRINS (*Figure 5*) was not replicated.

## Building a joint predictive model in GRINS and testing transferability to an independent sample

Finally, as a form of omnibus replication analysis, we tested the overall transferability of the GRINS findings by building joint models based on multiple domains of sleep metrics to predict case versus control status, using logistic regression. We used the SVD/PSC approach to derive components based on the sleep metrics which were selected by the effect size of group differences and the number of channels with significant differences in the GRINS dataset (*Figure 7*, right): FS and SS density, amplitude, ISA, duration, and chirp; SO density, duration, and slope; spectral power and net PSI. We first extracted components from each domain (spindle, SO, spectral power, and connectivity) via four separate SVD/PSC analyses, retaining components which (1) explained more than 1% of total variance and (2) displayed significant group differences (uncorrected p-value < 0.01). We then fitted a model to the GRINS data based on 12 selected components (four spindle, three SO, three spectral power, and two connectivity components). These 12 predictors achieved high classification (SCZ versus CTR) performance with AUC of 0.94, $R^2$ = 0.69 (corrected for overfitting using bootstrapped validation, although predictors were, of course, selected based on their univariate association with disease) (*Figure 10*, left). If taken separately, spindle-based components were the most critical for correct classification (AUC = 0.92, $R^2$ = 0.63 for spindle components versus SO components AUC = 0.87, $R^2$ = 0.5;

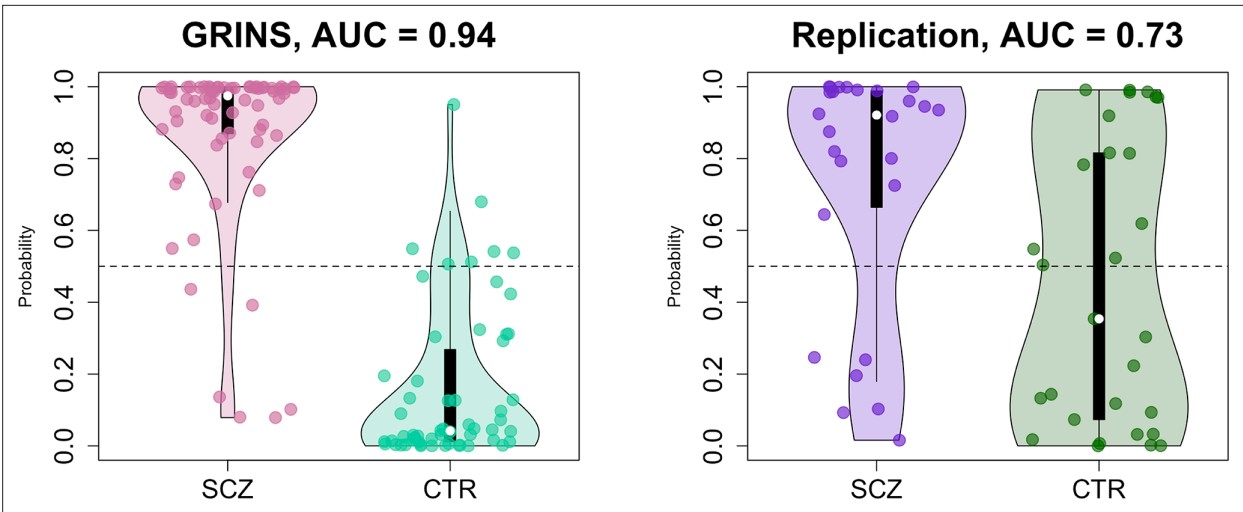

**Figure 10.** Predicting case/control status based on the sleep electroencephalogram (EEG): model building in GRINS and evaluation in an independent target sample. Two violin plots on the left illustrate the logistic regression probabilities for case/control labels for GRINS (training sample, left) and ESZ (independent target sample, right) using principal components based on spindle, slow oscillation (SO), spectral power and connectivity metrics. AUC – area under the ROC curve.

The online version of this article includes the following figure supplement(s) for figure 10:

**Figure supplement 1.** Schizophrenia (SCZ)/CTR prediction based on slow spindle (SS) density at P7 and fast spindle (FS) density at FC2.

PSD components AUC = 0.84, $R^2$ = 0.43; PSI components AUC = 0.74, $R^2$ = 0.22). Prediction based on spindle density alone (SS density at P7 and FS density at FC2) achieved an AUC of 0.8, $R^2$ = 0.35.

We evaluated the GRINS-derived model in the hdEEG replication cohort (26 cases, 29 controls, *Supplementary file 5*), to see whether it was broadly transferable to an independent and geographically/ethnically distinct population. We projected replication data into the GRINS-derived component space (see Materials and methods), that is, ensuring that 'component 4', for example, in the replication cohort indexed the same variability as in GRINS. Based on these projected components, we observed a full-model AUC of 0.73 in the replication sample (*Figure 10*, right). In contrast, prediction based on spindle density alone (SS density at P7 and FS density at FC2) achieved an AUC of only 0.64 in this independent sample (*Figure 10—figure supplement 1*).

## Discussion

We conducted a comprehensive study of NREM and wake abnormalities in SCZ using hdEEG recordings, basing our investigations on the previously reported spindle deficits (*Ferrarelli et al., 2010*; *Mylonas et al., 2020a*; *Wamsley et al., 2012*; *Manoach et al., 2014*) and extending to broader, multiple domains of NREM sleep macro- and microarchitecture, as well as wake ERPs implicated in thalamocortical functioning and SCZ (*Ferrarelli and Tononi, 2010*).

We provided statistically compelling evidence of a core set of spindle deficit in patients, consolidating prior reports based primarily on small sample sizes and varying methodologies. In addition, we demonstrated deficits in patients across multiple domains of NREM sleep including SO density and morphology, spectral power, SO/spindle coupling, and global connectivity, and replicated disrupted auditory processing during wake. Our findings in NREM were largely replicated in an independent and demographically distinct sample of patients and controls. A multivariate predictive model trained in GRINS using multiple NREM metrics was broadly transferable to the replication sample, and outperformed the model based on spindle density alone. Importantly, NREM sleep metrics were largely independent of one another, and did not correlate with wake ERP measures, suggesting multiple distinct thalamic and thalamocortical mechanisms underlying disease abnormalities. Taken together, these results support the utility of biomarkers that are based on a precise dissection of the sleep EEG, with the potential for risk and patient stratification that will ultimately map to distinct neurophysiological deficits.

Despite showing broadly comparable gross sleep macro-architecture in terms of stage duration and TST, we observed marked spindle impairments in SCZ, not associated with antipsychotic dose or adjunctive medication. Although we observed longer sleep latency in patients, it is not clear how generalizable this finding will be given the context of sleeping at the sleep center following ERP testing. Only a few studies have investigated SS and FS separately in SCZ (*Ferrarelli et al., 2010*; *Schilling et al., 2017*) despite multiple reports of their distinct topography, functional specificity, and SO coupling (*Purcell et al., 2017*; *Cox et al., 2017*; *Mölle et al., 2011*). In addition to finding that FS and SS density were largely uncorrelated and independently associated with SCZ, while reductions in the density and amplitude of both FS and SS were found in patients, the decrease in density was more prominent for FS, while the decrease in amplitude was more prominent for SS. Furthermore, despite observing similar average spindle deceleration (chirp) in both FS and SS, only FS showed more pronounced deceleration in patients. Although spindle chirp is a relatively novel metric, it has been shown that such intra-spindle frequency deceleration is a basic property of both SS and FS (*Andrillon et al., 2011*; *Schönwald et al., 2011*). While the mechanisms of how chirp is regulated are not established yet, our findings of altered chirp in SCZ should motivate future studies to investigate this question. In addition to differences in topography, one key distinction between SS and FS is in their coupling with SOs. One study demonstrated intact SO/FS temporal coupling in patients that was associated with overnight consolidation in motor task performance, but did not study SS (*Demanuele et al., 2017*). In GRINS, we observed reduced SO/spindle overlap in SCZ, but only for SS. Similarly, phase-frequency modulation was altered in SCZ only for SS. Spindles are generated in the TRN and modulated by thalamocortical connections. Interactions between different sub-regions of TRN and higher order or sensory thalamic nuclei have been described molecularly and functionally (*Li et al., 2020*; *Vantomme et al., 2020*), and may underlie frequency-dependent differences in spindle mechanisms. Alternatively, it is possible that unidentified spindle oscillators outside of TRN in the thalamocortical circuits contribute to SS and FS distinctions. Additional neurobiological studies are

warranted to clarify the source of these frequency differences in spindles. Regardless of their origin, SS density and amplitude had more evident associations with SCZ symptomatology (negative symptoms and disorganized thoughts) compared to FS, although this result has yet to be replicated in other samples. Considering that SS and FS properties were largely independent of each other, these distinct patterns of association with SCZ point to differences in their thalamocortical generation and propagation mechanisms and functional affiliations (e.g. relations to sleep-dependent memory consolidation).

Spectral power was reduced in SCZ in sigma and delta/theta frequencies. The former finding was expected due to spindle deficits and is consistent with previous studies with smaller samples (*Ferrarelli et al., 2007*; *Ferrarelli et al., 2010*; *Wamsley et al., 2012*; *Manoach et al., 2010*; *Manoach et al., 2014*; *Göder et al., 2006*). Spectral power in slow (0.5–1 Hz) or delta (1–4 Hz) bands has been used as a proxy for SO activity. Both unaltered and reduced slow/delta power have been reported in SCZ (*Ferrarelli et al., 2010*; *Göder et al., 2015*; *Wamsley et al., 2012*; *Manoach et al., 2010*; *Manoach et al., 2014*; *Göder et al., 2006*; *Keshavan et al., 1998*). Few studies considered other frequencies; two studies that did test multiple frequency bands reported decreased theta power in SCZ (*Manoach et al., 2010*; *Manoach et al., 2014*), which is consistent with our findings.

Prior reports considering discrete SO events (0.5–2 Hz) during NREM have also been inconsistent, with some studies supporting reduced density/amplitude of SO (*Keshavan et al., 1998*; *Hoffmann et al., 2000*; *Kaskie et al., 2019*; *Sekimoto et al., 2011*) but others showing unaltered SOs (*Ferrarelli et al., 2010*; *Mylonas et al., 2020a*; *Demanuele et al., 2017*; *Bartsch et al., 2019*). Underpowered samples, methodological differences, and/or medication effects may account for these apparent discrepancies (*Castelnovo et al., 2018*). Using the current GRINS sample – which is at least twice as large as most of the previous studies – we observed SO alterations in patients, including increased SO density, duration, and flatter slope. Only SO density differences replicated here, although it is plausible that the limited four-channel montage was a limitation. Caveats include that we only considered SOs during N2 sleep, potential methodological effects of thresholding choices, and associations between SO metrics and adjunctive medication use. Overall, although our findings suggest that some form of SO abnormality is plausible in SCZ, these effects are not as large or robust as the spindle deficit, despite the putative role of SO in orchestrating spindles (*Staresina et al., 2015*).

Global patterns of functional connectivity during N2 sleep were also affected in SCZ. While there is substantial evidence of abnormal waking connectivity as a hallmark of SCZ from fMRI (*Dong et al., 2018*; *van den Heuvel and Fornito, 2014*) and M/EEG studies (*Mackintosh et al., 2021*; *Maran et al., 2016*), connectivity findings in sleep are more scarce and restricted to coherence in spindle (*Wamsley et al., 2012*; *Sun et al., 2021*; *Markovic et al., 2020*) and slow/delta frequency bands (*Bartsch et al., 2019*), mostly pointing to reduced connectivity in patients, although there is also evidence of increased coherence (*Markovic et al., 2020*). To the best of our knowledge, only two prior studies have evaluated coherence at other frequencies during sleep. One study reported increased coherence across classical frequency bands in childhood onset SCZ (*Markovic et al., 2021*); another study investigated NREM coherence in beta and gamma frequencies in a very limited sample yielding null results (*Yeragani et al., 2006*).

The typically used magnitude squared coherence metric is very sensitive to volume conduction effects, precluding accurate estimation of true connectivity between channels (*Sakkalis, 2011*). Here, we used the PSI (*Nolte et al., 2008*), which mitigates volume conduction and shared reference issues. Summarized by channel, our results revealed a distinct pattern of directed connectivity across 3–20 Hz that was also seen in the ESZ replication dataset. We observed increased connectivity (higher absolute net PSIs) in SCZ for frequencies at 6–8 and 11–14 Hz. While increases in 6–8 Hz connectivity is partially congruent with a previous report (*Markovic et al., 2021*), hyperconnectivity for sigma appears to be the opposite of previous findings of decreased sigma coherence in SCZ. It is important to note, however, that these findings do not necessarily contradict one another, but rather indicate that such decrease in coherence is mostly driven by a zero-lagged connectivity component, which would be highly influenced by volume conduction and attenuated in PSI estimates. When pairwise sigma magnitude squared coherence was explicitly tested in GRINS, we found no evidence of decreased connectivity. Our results of sigma hyperconnectivity in SCZ as indexed by net PSI was replicated in the ESZ cohort, suggesting its robustness.

We clarified that alterations in SO, spectral power, and connectivity represented largely independent phenomena from the sleep spindle deficit: a model based on multiple domains better classified

disease status in an independent sample, compared to spindle density alone. Although still distant from diagnostic application in clinical settings, such results suggest that alterations in these aspects of sleep electrophysiology convey additional information on the neurobiological underpinnings of SCZ and may be helpful in stratifying patients with SCZ or distinguishing them from other neuropsychiatric conditions for which spindle density alterations have been reported (*Farmer et al., 2018*; *Ritter et al., 2018*).

In addition to spindle deficits, we report novel EEG alterations in SCZ patients including intra-spindle frequency change (chirp), overlap between slow spindles and SO, and increase in PSI indicative of functional hyperconnectivity. While the neurobiology of these metrics is yet to be studied extensively, there are clues to the function of these EEG features. For example, TRN neurons gradually hyperpolarize and fire with fewer action potentials upon repeated bursting which may underlie the chirp during the 'waning' of spindles (*Barthó et al., 2014*). Therefore, chirp could reflect intrinsic biophysical properties of TRN and thalamocortical neurons during this phase. Similarly, a recent study suggested that spindles nested within SO play a causal role in learning and are important for offline processing (*Kim et al., 2019*; *Latchoumane et al., 2017*), suggesting SCZ patients may be impaired during such reactivations of waking experience. The full relevance and functional implications of these findings in disease pathophysiology require further investigation, however.

With respect to the wake metrics, we broadly confirmed the previously reported ERP deficits in SCZ (*Erickson et al., 2016*; *Freedman et al., 2020*; *Thuné et al., 2016*; *Ferrarelli and Tononi, 2010*; *Atagun et al., 2020*; *Bramon et al., 2004*). Specifically, we observed impaired sensory gating (indexing deficits in filtering repetitive sensory stimuli), reduced MMN (reflecting abnormal automatic change detection process in auditory cortex), and decreased ASSR to 40 Hz auditory stimulation (associating with neural circuit abnormalities in this disorder) (*Erickson et al., 2016*; *Freedman et al., 2020*; *Thuné et al., 2016*). Leveraging the fact that our samples had both sleep and wake EEG recordings, we tested whether sleep spindle deficits and wake abnormalities were associated, as might be expected if they were to both depend on intact function of the same thalamocortical circuits (*Ferrarelli and Tononi, 2010*; *Pratt and Morris, 2015*). In sleep, the TRN is crucial for spindle generation, whereas in wake, the TRN is dynamically involved in controlling thalamic sensory gain and in gating less important sensory information (*Halassa et al., 2014*; *Wimmer et al., 2015*). While very few direct reports link the TRN to the neurophysiological bases of auditory EEG potentials listed above (*Lakatos et al., 2020*), there is a general agreement that efficient sensory processing inherently relies on precise coordination between thalamic nuclei and auditory cortex, and the TRN is a modulator between cortex and thalamus (*Ferrarelli and Tononi, 2010*; *Fernandez and Lüthi, 2020*; *Kimura et al., 2012*). Only a few studies to date have investigated the links between wake and sleep cortical alterations in SCZ. One recent study demonstrated that increased functional connectivity as measured by MRI between thalamus and somatosensory and motor cortex in wake inversely correlated with spindle density, suggesting deficient TRN inhibition as a plausible common mechanism (*Baran et al., 2019*). However, our findings did not point to strong linkages between individual differences in sleep metrics and wake auditory processing abnormalities. Furthermore, in response to a recently reported link between spindle deficits and auditory hallucinations in SCZ (*Sun et al., 2021*), we found that reduction in spindle density was not restricted to the subset of patients with auditory hallucinations. In summary, our present results suggest that the sleep EEG carries non-redundant information on thalamocortical circuits beyond the established ERP paradigms.

Our study is not without limitations. Although previous studies reported that spindle differences between SCZ and CTR were observed similarly during the first and the second naps/nights of recording (*Mylonas et al., 2020a*; *Mylonas et al., 2020b*), we could not directly test the first-night effect due to the absence of the adaptation night in our sample. We also could not directly assess the likelihood and impact of circadian disruption in SCZ, as our cohort was composed of inpatients whose sleep-wake patterns were constrained by hospital-imposed schedules. In our sample, almost all patients with SCZ were prescribed antipsychotic or adjunctive medication and some used more than one type of antipsychotic. While there is evidence of spindle deficits in unmedicated patients – and we also controlled for medication dosage – we cannot completely exclude medication effects in observed EEG alterations. Another potential caveat relates to difficulties interpreting functional connectivity at a sensor level. Some issues, for example, volume conduction, were partially mitigated by the choice

of connectivity metric, but accurate source reconstruction may facilitate interpretations with respect to precise cortical topography.

In summary, this work reinforces previous findings of a spindle deficit as a highly reproducible trait associated with SCZ pathophysiology across ethnically/racially distinct samples, as well as pointing to notable distinctions between SS and FS in patients. Although the spindle density deficit was the most marked, our analyses point to multiple domains of abnormalities of the NREM sleep and wake EEG that were independently associated with SCZ, potentially indexing different facets of thalamic and cortical dysfunction. Elucidating how these neurophysiological abnormalities co-exist and interact, and which aspects of SCZ pathophysiology they reflect is critical for a better understanding of SCZ etiology and heterogeneity. Analyzing these measures in genetically informative designs may reveal associations with disease risk, and lead to knowledge of the underlying molecular and neural circuits to reveal novel therapeutic targets to mitigate relevant abnormalities. Future research, including longitudinal characterization across the natural history of SCZ, will be needed to dissect which of these EEG features index different behavioral and cognitive manifestations of SCZ, in order to stratify patient populations, and understand their temporal development. Our findings support that sleep-based biomarkers have the potential to provide powerful phenotyping in animal, genetic, and clinical research aimed at developing novel therapeutic and preventive strategies for SCZ.

# Materials and methods
## Study design
### Participant cohort
The GRINS (Global Research Initiative of Neurophysiology on Schizophrenia) study is a part of the research collaboration between the Broad Institute/Harvard Medical School (USA) and Wuxi Mental Health Center (China). This is an ongoing study to assess sleep and wake neurophysiological parameters in SCZ patients. All sleep data collected by October 12, 2020, were included in the analyses. All participants were between 18 and 45 years with normal IQ range (>70) (*Jiang, 2006*; *Lian-e et al., 2010*). All SCZ patients were recruited from Wuxi Mental Health Center inpatients, and clinically diagnosed SCZ according to the Diagnostic and Statistical Manual of Mental Disorders (DSM-5). Control subjects were enrolled from the local community via advertisements and were required to have no current or a history of any mental disorder as well as no family history of psychiatric illnesses.

In addition, the following exclusion criteria were employed for all individuals: (1) received (modified) electroconvulsive treatment within the past 6 months; (2) any self-reported diagnosed sleep disorders including obstructive sleep apnea; (3) taking barbiturates; (4) any severe medical conditions, such as epilepsy or head injury; (5) having hearing impairment (threshold above 45 dB at 1000 Hz); and (6) in pregnancy or lactation.

All participants took part voluntarily and gave their informed consent. The study protocol complied with the Declaration of Helsinki and was approved by the Harvard TH Chan School of Public Health Office of Human Research Administration (IRB18-0058) as well as the Institutional Review Board of WMHC (WXMHCIRB2018LLKY003). Participant demographic information is shown in *Table 1*.

### Replication datasets
For the sleep EEG replication analyses, we compiled an independent dataset by pooling data from three previously reported studies: (1) 26 cases and 29 controls from the 'ESZ' dataset (*Mylonas et al., 2020a*; *Baran et al., 2019*), (2) 20 cases and 17 controls from the 'Lunesta' dataset (*Wamsley et al., 2012*; *Demanuele et al., 2017*; *Wamsley et al., 2013*), and (3) 11 cases and 13 controls from the 'GCRC' dataset (*Manoach et al., 2010*). See *Supplementary file 5* for demographic details.

Only the ESZ dataset contained high-density EEG with 54 channels in common with GRINS; most Lunesta participants had C3, C4, F3, F4, O1, O2, and Pz channels; the GCRC dataset only had C3, C4, O1, and O2. All channels were re-referenced to linked mastoids except the GCRC dataset where only contralateral mastoid referenced data was available.

Key spectral and spindle replication analyses were based on the limited montage available for all individuals; replication analyses requiring hdEEG (e.g. connectivity and PSC analyses) were necessarily restricted to the ESZ sample. Further, as we identified uncertainties over the EEG signal polarity

**Table 1.** Sample description.

| Sample characteristics | | SCZ | CTR |
|---|---|---|---|
| Sample size | | 72 | 58 |
| Sex, females (%) | | 25 (19%) | 22 (17%) |
| Age, years | | **35 ± 7*** | 32 ± 6.3 |
| Maternal education, individuals with higher than middle school education (%) | | 30 (23%) | 14 (11%) |
| Paternal education, individuals with higher than middle school education (%) | | 31 (24%) | 22 (17%) |
| | Positive scale | 15 ± 5.8 (mild) | |
| PANSS | Negative scale | 16 ± 5.7 (mild) | |
| | General scale | 33 ± 9 (mild) | |
| Duration of SCZ, years | | 11 ± 6.9 | |
| CPZ equivalent antipsychotic dose, mg | | 395 ± 222.9 | |
| Sleep macrostructure parameters | | | |
| Total time in bed, min (TIB) | | **532 ± 55.7 ‡** | 467 ± 44.5 |
| Total sleep time, min (TST) | | 378 ± 95.1 | 382 ± 63.1 |
| Sleep latency, min | | **50 ± 40.1 ‡** | 24 ± 27.3 |
| Wake after sleep onset, min | | 65 ± 44.1 | 49 ± 34.4 |
| Sleep efficiency (TST/TIB) | | **73 ± 14.1†** | 82 ± 10.9 |
| Sleep efficiency (TST/sleep onset-sleep offset period) | | 85 ± 10.7 | 88 ± 8.8 |
| Duration N1, min (%) | | 42 ± 26.2 (11%) | 32 ± 17.3 (9%) |
| Duration N2, min (%) | | 186 ± 69.8 (**50%***) | 204 ± 48.2 (53%) |
| Duration N3, min (%) | | 76 ± 49.8 (21%) | 80 ± 31 (21%) |
| Duration REM, min (%) | | 66 ± 37.6 (17%) | 65 ± 25.1 (17%) |
| REM latency after sleep onset, min | | 121 ± 55.2 | 115 ± 59 |
| Number of cycles | | 4 ± 1.4 | 4 ± 0.7 |
| Cycle length, min | | 99 ± 28.3 | 104 ± 21.9 |

* $p < 0.05$.
† $p < 0.01$.
‡ $p < 0.001$.

in GCRC dataset, this dataset was not included in polarity-dependent SO and coupling replication analyses.

The three datasets have previously reported spindle deficits in patients (see the references above). However, these three datasets have not previously been combined, and the comprehensive set of microarchitecture measures employed here has not been consistently applied across all studies. For example, (1) spectral power analysis was limited to band-specific analyses in GRRC and Lunesta, (2) only FS were measured, (3) spindle chirp was not assessed, and (4) and only one low-density study considered connectivity (measured as coherence). Nonetheless, given the extant literature, we present these replication samples not to address the general hypothesis of whether or not there is altered spindle activity in SCZ – as that would be circular – but rather to provide a methodologically integrated analysis of the specific sleep EEG metrics tested in GRINS.

## Behavioral and clinical data collection

All included participants completed three visits: one visit (1) to determine the eligibility, (2) the clinical assessment and recording demographic and other medical information, and (3) an overnight EEG recording visit, comprising an ERP session in the evening, followed by the sleep EEG.

Diagnoses of SCZ were confirmed using the Structured Clinical Interview for DSM Disorders (SCID) (*Zhou et al., 1997*; *Phillips et al., 2009*). Control subjects were not diagnosed with any major mental disorder and screened by a psychiatrist. General medical and clinical information collection were conducted by a full-time researcher, while PANSS (*He, 1999*) and SCID were carried out by trained psychiatrists in the Wuxi Mental Health Center within a week from the date of the sleep EEG recording. Intra-class correlation coefficients for PANSS ranged from 0.86 to 0.94, demonstrating excellent inter-rater reliability.

## EEG data acquisition

Participants were introduced to the EEG recording process and environment in advance, including the sleep ward where recordings occurred, the electrode cap and conductive paste, to acclimate them to the novel setting and procedures. All participants were asked to take a bath the night before the scheduled recording and wash their hair. EEG recordings were performed in a room at the sleep medicine center . EEG was recorded continuously from a customized 64-channel EasyCap using the BrainAmp Standard recorder (Brain Products GmbH, Germany) at the sampling rate of 500 Hz. The left clavicle electrode served as ground and FPz as the recording reference. The vertical electrooculogram recording electrode was placed at the lower orbit of the left eye, and the horizontal electrooculogram recording electrodes were placed 1 cm from the outer corner of both eyes. Electrode impedances were kept below 10 kOhm. Overnight sleep recordings were monitored by a researcher in an adjacent room; recordings were typically stopped after the participant was fully awake the next morning. While all subjects had the same sleep opportunity, in a minority of cases recordings were truncated prior to final wake, for logistical reasons dictated by the hospital schedule for the sleep ward (16 SCZ patients and 20 CTR, $p = 0.049$ for the group difference in rate of truncated studies). This occurred only in the final portion of the recording, meaning that participants with truncated recordings in fact tended to have slightly longer TSTs ($p = 0.002$). Truncation status was not associated with the NREM microarchitecture features which were shown to be associated with SCZ in this study (data not shown).

## Wake ERP paradigms

Wake EEG recording included three auditory ERP paradigms: dual-click paradigm for the P50 sensory gating, 40 Hz auditory click stimulation for the ASSR, and passive oddball paradigm for the MMN, with a 10 min break after each task.

The sensory gating paradigm followed established methods (*Hall et al., 2006*; *Hall et al., 2011*; *Hall et al., 2015*). Briefly, sensory gating ERP was elicited using the dual-click paradigm (S1 and S2 clicks); 160 paired-identical click stimuli (5 ms duration; 2 ms rise/fall period; 500 ms inter-click interval; 10 s inter-trial interval) were presented in four blocks (40 paired-click stimuli/block). Each block was separated by a 1 min break.

The ASSR paradigm followed our previous studies (*Hall et al., 2015*; *Zhou et al., 2018*). Briefly, an ASSR of gamma oscillation was elicited by a 40 Hz click stimulation paradigm (150 trains of 1 ms white noise clicks, 500 ms duration, 1100 ms stimulus onset asynchrony, 40 Hz stimulation rate) (*Spencer et al., 2008*).

Third, the MMN paradigm followed our established methods (*Higgins et al., 2021*). Briefly, a passive auditory oddball (in response to duration deviants) paradigm was employed. A total of 1200 stimuli consisted of 85% standard tones (1000 Hz, 100 ms) and 15% deviant tones (1000 Hz, 150 ms) with an inter-stimulus interval of 200 ms, and stimulus-onset-asynchrony 300 ms after standards and 350 ms after deviants. Stimuli were presented using the Presentation software (Neurobehavioral Systems, NBS) and foam insert earphones. During this task, participants were instructed to watch a peaceful silent cartoon video clip.

## Sleep EEG analysis

### Staging and pre-processing

All steps of sleep EEG data processing were performed using Luna, v0.25.5 (http://zzz.bwh.harvard.edu/luna/), an open-source package developed by us (SMP). EEG channels were re-referenced to linked mastoids, downsampled to 200 Hz and bandpass filtered between 0.3 and 35 Hz. Sleep stages were assigned (30 s epochs) through manual staging performed by a certified polysomnographic technician using standard AASM criteria, based on C4-M1, F4-M1, O2-M1, all EOG, and EMG channels (*Berry, 2015*).

Separately for each sleep stage, an automatic procedure to detect outlier epochs was employed to remove epochs likely containing artifacts. First, we identified channels with gross and persistent artifact, which were dropped and subsequently interpolated, using spherical spline interpolation (*Perrin et al., 1989*). Specifically, a channel was flagged as bad if over 30% of its epochs were more than 2 SD from the mean of all channels, for any of the three Hjorth parameters, activity, mobility, and complexity (*Hjorth, 1970*), that is, involving comparison within epoch across channels.

Second, we identified outlier epochs compared to all other epochs for all EEG channels, using the same Hjorth criteria but with a more stringent 4 SD threshold; further, all epochs with maximum amplitudes above 500 μV, or with flat or clipped signals for more than 10% of the epoch, were also marked as outliers and interpolated. Neither percentage of interpolated epochs nor the total number of interpolated channels differed significantly between the groups (p = 0.35 and p = 0.19, respectively).

Finally, separately for each channel, we flagged outlier epochs using a 4 SD threshold for the same set of Hjorth parameters (i.e. within-channel and within-epoch comparisons). Any epoch flagged at this stage for at least one channel was dropped from analysis for all channels, such that the final analytic dataset comprised the same set of epochs for all channels. This last step was repeated once more to ensure that there were no gross outlier epochs left.

Overall, this procedure removed an average of 7.4% epochs per individual in SCZ group and 8.5% epochs in CTR group. Although the proportion of removed epochs differed between diagnostic groups (p = 0.01), the total number of N2 epochs used in the final analyses was similar between groups (353±143 [21–758] epochs in SCZ and 374±89 [206–633] epochs in CTR, p-value = 0.16). Cleaned data were visually inspected and four subjects (all SCZ) were excluded due to persistent and severe line noise artifacts across most of channels for most of the night.

### Spindle detection

Given prior evidence for distinct topographies, functional specificity, and SO coupling properties of 'FS' and 'SS' spindles (*Purcell et al., 2017*; *Cox et al., 2017*; *Mölle et al., 2011*), spindles were detected via a wavelet method as previously described (*Purcell et al., 2017*; *Djonlagic et al., 2021*) using center frequencies of 11 and 15 Hz corresponding to SS and FS, respectively. These center frequencies were chosen to minimize the overlap between the detected SS and FS and maximize their differences in topography (frontal versus central) and coupling with SOs (see *Figure 2—figure supplement 3*).

Specifically, based on temporally smoothed (window duration = 0.1 s) wavelet coefficients (from a complex Morlet wavelet transform), putative spindles were identified as intervals exceeding (1) 4.5 times the mean for at least 300 ms and also (2) 2 times the mean for at least 500 ms. Intervals over 3 s were rejected; consecutive intervals within 500 ms were merged (unless the resulting spindle was greater than 3 s).

Finally, putative spindles were discarded if the relative increase in non-spindle bands activity (delta, theta, and beta) was greater than the relative increase in spindle frequency activity (i.e. relative to all N2 sleep), thereby ensuring putative spindles preferentially reflect sigma-band activity, and not general increases in signal amplitude.

For spindles that passed QC, we computed spindle density (count per minute), amplitude, ISA (sum of the normalized wavelet coefficients), duration, observed frequency, and chirp (intra-spindle frequency change). Chirp was based on the distribution of intervals of zero-crossings, computing the difference in the implied frequency between the first versus the second half of the spindle.

## SO detection

The EEG data were bandpass filtered between 0.3 and 4 Hz and zero-crossings of the resultant time-series were identified. Putative SOs were detected if the following temporal criteria were satisfied: (1) a consecutive zero-crossing leading to negative peak was between 0.3 and 1.5 s long; (2) a zero-crossing leading to positive peak were not longer than 1 s.

We employed two parallel approaches to amplitude filtering, following *Djonlagic et al., 2021*. Using an absolute threshold, only SOs with amplitude of the negative peak larger than –40 µV and amplitude between the positive and negative peak larger than 75 µV were identified as the final set of SOs. Using an adaptive/relative threshold (our default), only SOs with negative peak and peak-to-peak amplitudes greater than twice the mean (for that individual/channel) were selected.

Density of SO (count per minute) as well as the mean amplitude of the negative peak, peak-to-peak amplitude, duration, and the upward slope of negative peak were computed for each channel.

## SO/spindle temporal coupling

We identified SS and FS that overlapped with a detected SO (in the same channel); the proportion of spindles that overlapped an SO was recorded as one gross metric of SO/spindle coupling ('overlap'). To more precisely quantify the temporal coupling of spindle, we estimated the instantaneous phase of the SO based on a filter-Hilbert method (bandpass filtering between 0.5 and 4 Hz), and recorded the SO phase at the spindle peak (the point of maximal peak-to-peak amplitude). For each channel, we computed the (circular) mean SO phase at spindle peak; further, to assess the consistency of coupling, we computed the inter-trial phase clustering metric (the 'magnitude' metric). As previously described (*Djonlagic et al., 2021*), we randomly shuffled (10,000 times) the location of spindle peaks to obtain the null distribution of the overlap and magnitude metrics, in a manner that preserved the overall number of spindles, SO and (for the magnitude metric), also the gross overlap between SO and spindles. In this way, we controlled for the differences in coupling which could arise from group differences in spindle and SO density. The final metrics were the Z-scores for overlap and magnitude coupling metrics, that is, normalizing the observed statistic by the mean and SD observed under the null.

## Spindle instantaneous frequency estimation and SO/spindle phase/frequency modulation

We used a filter-Hilbert method (bandpass filtering ±2 Hz around the mean observed spindle frequency for that individual/channel) to provide an estimate the instantaneous spindle frequency for every sample point. Using this, we summarized the frequencies for each spindle quintile, averaging over all spindles for that individual/channel and target spindle frequency. We additionally averaged spindle frequency with respect to SO phase (in eighteen 20-degree bins). Finally, we averaged frequencies jointly by spindle quintile ('progression') and SO phase bin.

To quantify the extent to which spindle frequency varied as a function of SO phase for a particular individual/channel and spindle target frequency, we estimated the circular-linear correlation between SO phase and average frequency (based on the same 18 SO phase bins).

## Spectral power and functional connectivity

Spectral power was estimated for the frequency range between 0.5 and 20 Hz using Welch's method. For each 30 s epoch, power spectra were estimated by applying the fast Fourier transform within 4 s segments (0.25 Hz spectral resolution) windowing with a Tukey (50%) taper; segments overlapped by 50%, and then averaging power across all segments per epoch. Subsequently, epoch-wise power was then averaged across all epochs.

To evaluate connectivity between channels, we used the PSI (*Nolte et al., 2008*), which is based on the imaginary part of coherence to minimize spurious connectivity due to volume conduction. The PSI is based on how the phase delay between two signals changes as a function of frequency, and is a directional metric that can be interpreted as indicating the 'flow of information', whereby a positive PSI value for channel pair (A, B) is consistent with A temporally preceding B, whereas a negative value implies that A lags B. We estimated the PSI for all channel pairs for 10 min (20 epochs) of N2 sleep, randomly selected for each subject. For each epoch, the PSI was estimated from 3 to 20 Hz in 1 Hz intervals, using a window of 5 Hz (e.g. slope of phase difference between two signals for 3 Hz will be estimated over a frequency range from 0.5 to 5.5 Hz). The PSI was calculated for 4 s segments, using

a 2 s interval with 50% overlap. For each epoch, the PSI values were normalized based on the SD of the PSI as described by *Nolte et al., 2008*. Final PSI values were then obtained by averaging the normalized values over all 30 s epochs.

As well as pairwise metrics, we also computed the net PSI for each channel, based on the sum of pairwise PSI values involving that channel, to indicate whether a single channel is, on balance, a 'sender' versus 'recipient' of information from other channels.

## Dimension reduction (PSC analysis)

Many sleep EEG metrics are estimated not just for 57 channels but also across a broad range of frequencies: for example, PSD was estimated for 79 bins (0.5–20 Hz in 0.25 Hz bins) yielding a total of 4503 highly inter-correlated metrics. To summarize patterns of individual differences across such sets of metrics, we employed dimension reduction using SVD/principal components analysis (PCA). We have referred to this approach in this context as PSCs analysis (*Djonlagic et al., 2021*). For a feature matrix $\mathbf{A}$ (rows corresponding to $N$ individuals, columns to the $M$ features), the SVD is defined as $\mathbf{A} = \mathbf{U\Sigma V^T}$, where $\Sigma$ is a diagonal matrix. Here, columns of $\mathbf{U}$ ($N \times M$, left singular vectors) contain the set of orthogonal component scores; the diagonal elements of $\Sigma$ ($M$ singular values) indicate the variance accounted by each component; the $\mathbf{V}$ matrix ($M \times M$, right singular vectors) map the components to the individual features (columns of $\mathbf{A}$), thereby indicating the topographical and frequency specificity of each component. In practice, select a subset $K << M$ components. Specifically, we selected components which explained at least 1% of total variance (11 principal components [PCs] in spectral power and 22 for net PSI analyses): for these components, the corresponding column of $\mathbf{U}$ constituted the new metric for the $N$ individuals.

## ERP analysis

### Sensory gating metrics

Signal processing was performed off-line using BrainVision Analyzer 2.2 (Brain Products, Germany) software and blind to group membership. Continuous EEG from each channel was segmented (−100 to 400 ms), creating 320 sweeps and a 1 Hz high-pass filter (24 dB/oct) was applied. Segments were then baseline-corrected using the pre-stimulus interval. An automatic artifact rejection procedure identified and rejected any sweeps 50 μV between 0 and 75 ms post-stimulus (to capture blinks and other slow-wave activity). Artifact-free sweeps were averaged for each of the five blocks for S1 and S2 separately. Average waveforms were then digitally filtered using a 10 Hz high-pass filter (24 dB/oct) with zero phase shift, and a 7-point moving average was applied twice (smoothing). At each site, the S1 response was identified as the most prominent peak in the 40–85 ms post-stimulus window. The preceding negative trough was used to calculate the S1 amplitude. For the S2 response, the positive peak with latency closest to that of the S1 peak was selected. The S2 amplitude was determined in the same way as the S1 amplitude. P50 suppression was calculated as the ratio (S2/S1) × 100 (*Hall et al., 2006*, *Hall et al., 2011*). A higher ratio reflects more impairment in sensory gating.

### ASSR metrics

Signal processing was performed off-line using Brain Vision Analyzer software and blind to group membership. EEG data were downsampled to 256 Hz, re-referenced off-line to the linked mastoids, and filtered between 0.1 and 50 Hz. Single-trial segments were extracted (from −500800 ms), baseline-corrected relative to the 500 ms pre-stimulus interval, eye-blink-corrected using Analyzer's default setting (*Gratton et al., 1983*) and artifact rejected if activity exceeding >100 μV. Gamma phase locking (γPL) and evoked power (γEP) at each site were calculated on wavelet coefficients obtained from Morlet wavelet transformation of the segmented data (representing the 1–50 Hz frequency range, with a total number of 50 frequency layers using a Morlet parameter of 10). γPL quantifies consistency of oscillatory phase across individual trials, ranging from 0 (purely non-phase-locked activity) to 1 (fully phase-locked activity). Phase locking and evoked power were computed by averaging across the 36 add symbol 46 Hz wavelet frequency layers at 20–520 ms window.

### MMN metrics

Signal processing was performed off-line using Brain Vision Analyzer software. EEG data were downsampled to 256 Hz, re-referenced off-line to the linked mastoids, filtered with high cutoff at 20 Hz,

segmented by stimulus marker from –100 to 400 ms. Segments were baseline-corrected using –100 to 0 ms pre-stimulus period and eye-blink-corrected using established method (*Gratton et al., 1983*). Artifact rejection for individual channels was performed and a given segment was rejected if the voltage gradient exceeded 50 µV/ms, amplitude was ±100 µV, or the signal was flat (<0.5 µV for >100 ms). MMN score was generated by subtracting ERP waveforms between standard and deviant stimulus. MMN amplitude was measured as the peak amplitudes between 120 and 250 ms.

We excluded one subject (SCZ), 16 subjects (13 SCZ and 3 CTR), and 27 subjects (14 SCZ and 13 CTR), from MMN, ASSR, and P50 sensory gating analyses, respectively, due to persistent line noise artifacts, excessive head movements, excessive ocular artifacts (for P50), or artifact-free segments less than 80% of the total segments. ERP data of those individuals who were excluded from ERP analyses were visually inspected to confirm the exclusion of the data.

## Statistical analysis

### Group differences and multiple test correction

Group differences were estimated using a logistic regression model, separately for each channel and including age and sex as covariates.

$$\text{logit (DISORDER)} \sim \text{EEG metric} + \text{AGE} + \text{SEX} + \text{error}$$

where 'EEG metric' and AGE are continuous variables, DISORDER (SCZ or CTR) and SEX (male or female) are binary predictors. Outlier EEG metric values (>3 SD from the mean) were removed prior to analysis.

To control for the multiplicity of tests across EEG channels, we employed a permutation-based approach to estimate adjusted, empirical significance values. Specifically, for each metric we generated a null distribution of asymptotic p-values by randomly permuting subjects' group (age and sex was fixed with an individual) and fitting the above model on the null data. For each of 3000 null replicates generated, we recorded the minimum null p-value (for the EEG metric) across all channels; adjusted empirical p-values were, for channel $C$, calculated as $(R_C + 1)/(3000 + 1)$, where $R_C$ was the number minimum null p-values that were lower than the original observed p-value for that channel.

All replication results were based on nominal, uncorrected asymptotic p-value (note that most replication comparisons involved only four EEG channels). Likewise, for the secondary and exploratory analyses of intra-spindle frequency modulation, we present nominal asymptotic results, using thresholds as indicated in the corresponding figure or table legends.

To control for variability due to technical and procedural differences between the three cohorts combined for the replication analyses, we used a mixed effect model with disorder, age, and sex as fixed effects and the study dataset as a random effect to test significance of group differences when two or more replication cohorts were combined (spindle, SO, coupling, and spectral power metrics).

### Cross-metric associations

Association between sleep metrics and wake EEG was investigated using Pearson's correlation at the Cz channel. Prior to that, we regressed out the effects of disorder, age, and sex, removed 3SD outliers.

For within-SCZ analyses of clinical measures (illness duration, PANSS scores; *Supplementary file 2*) and total antipsychotic dose as equivalent to 100 mg of chlorpromazine (*Supplementary file 4*), we used a multiple linear regression model.

$$\text{EEG metric} \sim \text{Clinical metric} + \text{AGE} + \text{SEX} + \text{error}$$

### Joint models: prediction of disease status given multiple EEG metrics

To assess the relative independence of different sleep EEG metrics in their association with SCZ, and to more globally quantify the variance explained by these metrics, we again used logistic regression with disease status as the dependent variable, but with multiple sleep EEG predictors.

To create the predictor variables, we first employed a round of dimension reduction to summarize domains of metrics (each metric measured for 57 EEG channels) into an order-of-magnitude smaller set of PCs that summarized patterns of individual differences. We generated PCs based on three groups of measures: (1) spectral power metrics across channels and frequency bins (based on 4503 metrics in total), (2) net PSI connectivity metrics across channels and frequency bins (based on 1026

metrics), (3) spindle parameters, based on density, amplitude, duration, ISA, and chirp across channels, for both FS and SS (570 metrics in total), and (4) SO parameters including density, duration, and slope (171 metrics in total). We did not include ERP metrics due to the absence of these data in the independent sample. Coupling parameters were not included because they expressed less extensive alterations compared to spindles and SO characteristics.

PCs were obtained by applying SVD separately to the three matrices of features described above (first mean-centering columns). From each analysis, PCs were selected based on the following criteria: (1) >1% total variance explained, (2) exhibiting a significant group difference (in GRINS) with a nominal $p < 0.01$. Overall, of the 54 components that accounted for >1% of the variance, 12 were significantly associated with SCZ (when less than one expected by chance). Of the 12 PCs, four were derived from spindle parameters; three from SO parameters; three from spectral power, and two from net PSI/connectivity.

Prior to fitting the logistic model, we regressed the effects of age and sex from each PC. Based on the logit-scale predicted values from the GRINS model, we derived the posterior probability of being SCZ versus CTR for each individual in GRINS. To quantify model performance in terms of case/control discrimination, we estimated the area under the ROC curve (AUC), using the 'rms' R package to correct for overfitting via a bootstrapping approach (*Efron and Tibshirani, 1994*).

To determine whether this model was broadly transferable to other populations, we applied the GRINS-derived model to generate posterior probabilities for the ESZ data (26 cases, 29 controls). This analysis could not include Lunesta and GCRC replication samples, as only ESZ had hdEEG comparable to GRINS.

To ensure the correspondence of PCs in GRINS and ESZ datasets, we used only the subset of 54 EEG channels that were common to both studies for this analysis. Further, we projected the ESZ features (spectral power, spindle and SO parameters, and net PSI metrics) into the twelve GRINS-derived components. Specifically, if the GRINS feature matrix $\mathbf{A}$ is decomposed by SVD as $\mathbf{A} = \mathbf{U\Sigma V^T}$, where the singular vectors (i.e. components) are the columns of $\mathbf{U}$, then for ESZ feature matrix $\mathbf{B}$, singular vectors can be calculated using $\mathbf{\Sigma}$ and $\mathbf{V^T}$ from the GRINS analysis, namely $\mathbf{U_B} = \mathbf{BV^T\Sigma^{-1}}$. As above, we regressed out any effects of age and sex in the projected PCs. Finally, using the original GRINS model, we estimated the predicted case probabilities for each ESZ individual, given these 12 projected PCs (*Figure 10*).

## Acknowledgements

We thank Stephanie A Marvin for manual scoring of all GRINS sleep studies.

## Additional information

### Funding

| Funder | Grant reference number | Author |
|---|---|---|
| Stanley Center for Psychiatric Research, Broad Institute | | Jen Q Pan<br>Hailiang Huang<br>Shaun M Purcell<br>Mei-Hua Hal |
| National Institute of Mental Health | R01 MH115045-01 | Jen Q Pan |
| National Institute of Neurological Disorders and Stroke | NS108874 | Jen Q Pan |
| National Institute of Mental Health | R01MH118298 | Jen Q Pan |
| National Institute of Mental Health | R03 MH108908 | Shaun Purcell |
| National Heart, Lung, and Blood Institute | R01HL146339 | Shaun Purcell |

| Funder | Grant reference number | Author |
| --- | --- | --- |
| National Heart, Lung, and Blood Institute | R01 HL146339 | Shaun Purcell |
| National Institute on Minority Health and Health Disparities | R21 MD012738 | Shaun Purcell |
| National Institute of Mental Health | R01MH092638 | Dara Manoach |
| National Institute of Mental Health | UG3 MH125273 | Dara Manoach |
| National Institute of Mental Health | K23MH118565 | Michael Murphy |
| Brain & Behavior Research Foundation Young Investigator and the Zhengxu and Ying He Foundation awards | | Hailiang Huang |
| Wuxi Mental Health Center | | Zhenhe Zhou |
| National Institute on Minority Health and Health Disparities | R21 HL145492 | Shaun M Purcell |
| Top Talent Support Program for Young and Middle-aged People of Wuxi Health Committee | HB2020077 | Jun Wang |

The funders had no role in study design, data collection and interpretation, or the decision to submit the work for publication.

## Author contributions

Nataliia Kozhemiako, Formal analysis, Investigation, Validation, Visualization, Writing – original draft, Writing – review and editing; Jun Wang, Data acquisition, Data curation, Formal analysis, Investigation, Resources, Writing – review and editing; Chenguang Jiang, Data acquisition, Data curation, Formal analysis, Investigation, Project administration, Writing – review and editing; Lei A Wang, Formal analysis, Investigation, Writing – review and editing; Guanchen Gai, Kai Zou, Zhe Wang, Xiaoman Yu, Guoqiang Wang, Data acquisition, Data curation, Writing – review and editing; Lin Zhou, Data curation, Resources; Shen Li, Robert Law, James Coleman, Formal analysis, Writing – review and editing; Zhenglin Guo, Project administration, Writing – review and editing; Dimitrios Mylonas, Data curation, Validation, Writing – review and editing; Lu Shen, Shengying Qin, Data curation, Writing – review and editing; Shuping Tan, Data curation, Methodology, Study design, Writing – review and editing; Hailiang Huang, Conceptualization, Writing – review and editing; Michael Murphy, Methodology, Writing – review and editing; Robert Stickgold, Conceptualization, Methodology, Validation, Writing – review and editing; Dara Manoach, Conceptualization, Validation, Writing – review and editing; Zhenhe Zhou, Conceptualization, Data curation, Project administration, Study design, Supervision, Writing – review and editing; Wei Zhu, Conceptualization, Data curation, Project administration, Study design, Writing – review and editing; Mei-Hua Hal, Conceptualization, Formal analysis, Methodology, Study design, Supervision, Writing – review and editing; Shaun M Purcell, Conceptualization, Formal analysis, Funding acquisition, Investigation, Methodology, Resources, Software, Study design, Supervision, Writing – original draft, Writing – review and editing; Jen Q Pan, Conceptualization, Data curation, Funding acquisition, Investigation, Project administration, Resources, Study design, Supervision, Writing – original draft, Writing – review and editing

## Author ORCIDs

Nataliia Kozhemiako http://orcid.org/0000-0002-6450-4959
Shengying Qin http://orcid.org/0000-0002-8458-5960
Zhenhe Zhou http://orcid.org/0000-0002-1334-8335
Shaun M Purcell http://orcid.org/0000-0002-7402-5812
Jen Q Pan http://orcid.org/0000-0003-0767-086X

### Ethics

Human subjects: All participants took part voluntarily and gave their informed consent. The study protocol complied with the Declaration of Helsinki and was approved by the Harvard T.H. Chan School of Public Health Office of Human Research Administration (IRB18-0058) as well as the Institutional Review Board of WMHC (WXMHCIRB2018LLKY003).

### Decision letter and Author response

Decision letter https://doi.org/10.7554/eLife.76211.sa1
Author response https://doi.org/10.7554/eLife.76211.sa2

## Additional files

### Supplementary files

• Transparent reporting form

• Supplementary file 1. Group differences in EEG parameters between SCZ and CTR.

• Supplementary file 2. Prescribed medications and their associations with EEG metrics within the SCZ sample.

• Supplementary file 3. Significance of group differences between SCZ and CTR adjusted for medication.

• Supplementary file 4. Association between clinical variables and EEG metrics in the SCZ sample.

• Supplementary file 5. Demographic characteristics of the independent samples.

• Supplementary file 6. Replication analysis for spindle, slow oscillation and coupling metrics.

### Data availability

Anonymized individual-level data for the derived EEG metrics, disease status, and demographics are available in a Dryad archive (doi:10.5061/dryad.j0zpc86h4). R scripts to reproduce the key figures are available at this GitLab repository: https://gitlab-scm.partners.org/zzz-public/grins/https://gitlab-scm.partners.org/zzz-public/grins/ (copy archived at swh:1:rev:6e5a8a8586f1a8b2a0996cf4ed55fff1e31c3333).

The following dataset was generated:

| Author(s) | Year | Dataset title | Dataset URL | Database and Identifier |
|---|---|---|---|---|
| Kozhemiako N, Wang J, Jiang C, Wang L, Gai G, Zou K, Wang Z, Yu X, Guo Z, Huang H, Zhou Z, Zhu W, Hall M, Purcell S, Pan J | 2022 | NREM sleep EEG and wake ERP summary data - the first wave of the Global Research Initiative on the neurophysiology of schizophrenia (GRINS) | https://doi.org/10.5061/dryad.j0zpc86h4 | Dryad Digital Repository, 10.5061/dryad.j0zpc86h4 |

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
