## [Editor Report]

This study, one of the largest of its kind, replicates previous findings regarding the impairment of sleep rhythms in patients with schizophrenia relative to healthy controls. Specifically, sleep spindles, which constitute a hallmark of non-Rapid Eye Movement sleep, are less frequent in people with schizophrenia and several other sleep features were also affected. Overall, this study provides evidence that brain dynamics during sleep are promising biomarkers for the diagnosis and the prevention of schizophrenia.

---

## [Decision Letter]

**Decision letter after peer review:**

Thank you for submitting your article "Non-rapid eye movement sleep and wake neurophysiology in schizophrenia" for consideration by *eLife*. Your article has been reviewed by 3 peer reviewers, and the evaluation has been overseen by a Reviewing Editor and Christian Büchel as the Senior Editor. The reviewers have opted to remain anonymous.

1) This study replicates previous findings regarding the association between sleep rhythm and schizophrenia in a large cohort of individuals. As the findings are not novel, the study must present solid and rigorous analyses to warrant publication. One critical aspect of the study is that sleep and circadian rhythms are often profoundly disrupted in patients with schizophrenia. The study thus needs to address the contribution of time of day, sleep homeostasis and light, among other external factors. Sleep-wake history and circadian rhythms were not taken into account neither, despite overwhelming evidence that these factors affect spindle dynamics in very profound ways.

2) Several methodological issues were raised. The range of frequencies that are analyzed should be broadened and not limited to the two ends (slow/fast) of the spectrum. Data should be analyzed in a standard way, especially for the topomaps: the study should show differences between the groups across electrodes for a frequency band, not just a single frequency bin.

3) The number of patients should be clarified, and the exclusion of subjects in some analyses should be always justified.

4) The manuscript should be somehow simplified, especially the supplementary information. Many supplementary figures do not add helpful information.

5) All sleep analyses were performed on N2 epochs. Analyzing N2/N3 combined and/or N2 and N3 separately would likely provide a better characterization of these sleep parameters. Furthermore, because the schizophrenia group had less stage 2 sleep than the control group, the same number of epochs should be used in the analyses of both groups.

6) Some of the limitations should be clearly stated, for example that no adaption night was included in the study.

*Reviewer #2 (Recommendations for the authors):*

In addition to the concerns raised in the public review (see weaknesses), a couple of additional points:

The authors should clarify why they did not perform an initial analysis targeting the entire spindle range, but instead decided to focus on slow spindles and fast spindles. Indeed, in their previous sleep study (Purcell et al., 2017, ref. 14) the same wavelet-based algorithm employed was used to initially target the whole spindle activity range and was centered at 13.5 Hz. Also, the choice of centering the analysis for slow spindles at 11 Hz and for fast spindles at 15 Hz is somewhat surprising, given that these frequency bins are usually considered the lowest and highest values respectively of the spindle activity. Furthermore, it appears that all sleep analyses were performed on N2 epochs. However, analyzing N2/N3 combined and/or N2 and N3 separately would likely provide a better characterization of these sleep parameters.

In previous work, the authors emphasize the role of sleep spindles in relation to cognitive function. It is therefore unclear why in the present study they did not examine the association of sleep metrics with cognitive parameters.

*Reviewer #3 (Recommendations for the authors):*

1) Why were 4 subjects removed for the primary analytic sample (page 7)?

2) The sample size in the abstract is a bit misleading. Please specific the number of schizophrenia participants and controls instead of writing 130 schizophrenia patients and control.

3) Figure 1: The authors only show topomaps at 3.25 Hz and 13.25 Hz in the topomaps because these frequencies show the maximal difference between the groups, but a difference at single frequency bin is not biologically meaningful and it is misleading. Please stick to the convention of showing differences between the groups across electrodes (topomap) for a frequency band. Same applies for the scatterplots.

4) The authors write: With respect to SO phase, both SS and FS tended to occur earlier in SCZ, albeit this effect was only observed at a few parietal channels (P5 and P7 for SS, max e.s.=-1.07 SD and P4, P6 for FS, max e.s.=-0.65 SD)." Only one channel is highlighted in Figure 3 for SS and one for FS. I would not draw conclusions based on the results of one channel.

5) What does chirp tell us about the neurobiology of the disorder? An additional way to characterize spindles doesn't necessarily tell us much unless the cellular/molecular mechanisms have been clarified.

6) For the PSI why are data presented frequency by frequency instead of averaging into bands? Again, showing the statistics for the most significant single frequency bin is misleading. I'm unsure how to read Figure 5 since from my understanding PSI is a measure between two channels but here each channel is shown with a single color. Perhaps I missed something here.

7) There are too many supplementary figures which aren't all helpful. For example suppl Figure 3 is not necessary. I can't find a reference for Supplementary Figure 1. Not sure what Supplementary Figure 2 is showing and how it relates to the two groups. Please consider carefully if all these figures are necessary

8) In the abstract the authors write "The main sleep findings were replicated in a demographically distinct sample, and a joint model, based on multiple NREM components, predicted disease status in the replication cohort." Prediction is a strong word given that the AUC is 0.64.

9) Because the schizophrenia group had less stage 2 sleep than the control group, the same number of epochs should be used in the analyses of both groups. Similarly, significantly more artifacts were removed from the Schizophrenia group which may introduce bias and reflect a problem with the automatic artifact removal. Any ideas why more artifacts were removed from the schizophrenia group?

10) The differential referencing of the GCRC data set will have a large impact on the data and you should consider excluding this data set if the goal is to directly compare these previously published data sets.

11) No adaptation night was included. This should be incorporated into the discussion as a limitation of the study.

12) What was the interval between the PANSS and the sleep EEG?

13) Was prior sleep controlled? The differences in slow oscillations could be due to differential sleep-wake history of patients versus controls.

14) The authors write that a minority of recordings ended prior during sleep. How many were these and did they differ between the two groups.

*Reviewer #4 (Recommendations for the authors):*

The two fundamental processes relevant for sleep regulation – sleep homeostasis and its circadian control were not considered in this study, which makes some of the central findings difficult to interpret.

For example, as well known slow wave activity is much greater in the early night as compared to morning hours. Yet, this highly significant dynamics was not taken into account. Conversely, the incidence of sleep spindles normally increases across the night, showing largely an opposite trends to slow-wave activity. These dynamics may meaningfully affect some of the metrics studied here, especially the coupling between slow oscillations and spindles. Some of the data presented in the paper are difficult to understand unless further information is provided on the dynamics of sleep oscillations across the night. At the very least I would recommend analysing separately the first and the last sleep cycle, but the authors may have other ways of addressing this important omission.

Further, as well known, patients with schizophrenia can manifest major circadian abnormalities in their mood and sleep behaviour. This is especially important given that the distribution of sleep stages and sleep oscillations, and specifically sleep spindles, are under a strong circadian control (as we know from forced desynchrony studies). Therefore, it seems essential to provide further information about circadian characteristics of sleep in the population studied, and the potential role of relevant factors, such as light levels, which are, surprisingly, never mentioned in this study. Please provide data on the levels of light the patients and controls were exposed to during the experiments. As I presume there is no data available on DLMO or body temperature, it is difficult to assess whether and to what extent the potential dysfunction of the circadian clock affects the results presented here. I would recommend providing more data on the time of sleep onset and morning awakening, relative to the light-dark cycle. Furthermore, it would be essential to provide further information on the season (winter, spring, summer, fall) when the data were collected.

---

## [Author Response]

1) This study replicates previous findings regarding the association between sleep rhythm and schizophrenia in a large cohort of individuals. As the findings are not novel, the study must present solid and rigorous analyses to warrant publication. One critical aspect of the study is that sleep and circadian rhythms are often profoundly disrupted in patients with schizophrenia. The study thus needs to address the contribution of time of day, sleep homeostasis and light, among other external factors. Sleep-wake history and circadian rhythms were not taken into account neither, despite overwhelming evidence that these factors affect spindle dynamics in very profound ways.

We agree and hope that this study provides a comprehensive analysis of the NREM sleep EEG in schizophrenia, including less-studied metrics such as intra-spindle frequency change.

With regard to circadian rhythms, we would like to point out that our SCZ cohort was composed of hospital inpatients, whose circadian rhythms and sleep patterns were strongly influenced (and regularized) by hospital-imposed schedules. As such, this cohort is not ideally suited to address the influence of circadian disruption as observed in community-based settings: this has been noted as a limitation in the Discussion on p. 32.

On the other hand, this regularization can also be seen as a strength of our study, inasmuch as profiles of NREM micro-architecture in SCZ are less likely to reflect nonspecific effects of generally disrupted sleep or sleep schedules. To confirm this analytically, we conducted additional analyses using data collected prior to the EEG night: i) a sleep habits questionnaire, ii) a two-week sleep journal and iii) bedtime and wake-up time for the night before the EEG. Analyses across all three data types congruently indicated that SCZ patients tended to go to bed earlier than control participants (all p-values < 10^-10^). Estimated total time in bed (TIB) was also longer in SCZ patients compared to CTR (all p-values < 10^-8^). Importantly, these results matched the group differences we originally reported for the EEG night (earlier bedtime, p = 7x10^-18^; longer TIB, p = 1x10^-10^) suggesting that the EEG night was representative of the regular sleep patterns in our SCZ cohort, characterized by earlier bedtimes and longer sleep latencies, but otherwise exhibiting sleep and stage durations broadly comparable with the CTR group. We added this information on p. 5.

Earlier bedtime and earlier wake-up time (the latter is only significant based on the sleep journal and the night before EEG recording) indicated earlier chronotype in SCZ patients. Similar to our report, Martin et al., 2005 showed earlier bedtime and longer TIB in SCZ. However, a more recent study found high inter-subject variability in SCZ which suggested two subgroups of SCZ patients exist with either earlier or later bedtimes compared to controls (Wulff et al., 2012). As noted, given that our cohort is composed of inpatients whose sleep/wake timing were strongly determined by hospital schedules, this study was not designed to address their natural circadian rhythm.

Next, we directly investigated whether bedtime or TIB prior or during the EEG night were associated with sleep microarchitecture metrics altered in SCZ in channels with the largest effect size. Except the SO density at O1 which was positively associated with TIB of the night before in both groups (p=0.041 in CTR and p=0.042 in SCZ), none of the tested sleep microarchitecture parameters (i.e. slow and fast spindle density, amplitude, ISA, FS duration, and chirp; SO duration, and slope; slow spindle overlap with SOs, SO coupling angle at slow and fast spindle peaks) displayed significant associations with TIB or bedtime congruently in SCZ and CTR groups. Furthermore, despite the revealed significant association in SO density at O1 , adding TIB of the night before as covariate could not explain the observed SCZ/CTR group differences in SO rate (original p=5.5x10^-5^ vs p= 0.004 controlling for TIB of the night before ).

To address concerns regarding the seasonal/light exposure effects, we used the R package ‘suncalc’ to estimate an exact time of sunrise on the day of the EEG, finding no significant difference between SCZ and CTR groups (p=0.48). Regarding artificial light exposure during the procedures prior to sleep, all participants followed an identical protocol in the same hospital sleeping ward precluding systematic differences that could impact sleep.

2) Several methodological issues were raised. The range of frequencies that are analyzed should be broadened and not limited to the two ends (slow/fast) of the spectrum. Data should be analyzed in a standard way, especially for the topomaps: the study should show differences between the groups across electrodes for a frequency band, not just a single frequency bin.

With regard to single frequency bin analyses, we would like to stress that our use of Welch's method significantly reduces the variance of spectral power estimates: power for each 30-second epoch is averaged over 14 overlapping 4-second windows (with a frequency resolution of 0.25 Hz per bin), and for a stage, power is again averaged over dozens or hundreds of epochs. As such, individually testing frequency bins across a full range, and using an approach to multiple testing that allows for highly correlated predictors (i.e. our use of permutation-based empirical significance estimates) is warranted.

Nonetheless, we agree that summarizing by classical frequency band is also a common approach to present findings in spectral power, and so we have included those analyses. Case/control differences in spectral power for classical frequency bands are now given in Figure 1 —figure supplement 1.

While bin-based and band-based results are congruent for σ band/range frequencies, our δ/theta (2-6 Hz) results are not well captured by classical band definitions. Author response image 1 clearly illustrates a physiological group difference in NREM power that does not respect the traditional (useful but ultimately arbitrary) convention of a precise and binary delineation of activity at 4 Hz. When effects do fall along classically-defined lines (e.g. σ), our bin-based analysis will also show this; when they do not (e.g. the 2-6 Hz result), a band-based approach runs the risk of false negatives. To avoid such misrepresentation, we kept the original way of reporting the power findings in the main text.

We would also like to further clarify that our analysis of spectral power was performed across all frequency bins from 0.5 to 25 Hz. We show raw data plots for the most significant metrics only, as it is not feasible to show plots for all metrics and all channels: as we employed rigorous multiple-test correction, this approach does not unduly capitalize on chance.

**Author response image 1. sa2fig1:** 

Finally, we note that the principal spectral component (PSC) approach also provides a complementary mode of analysis that does not rely on individual frequency bins. However, unlike band-based approaches, PSC forms linear combinations across all frequency bins (and channels) in a data-driven manner (which also implicitly indexes other parameterizations of EEG power spectra not reflected by band-based analysis, e.g. of spectral intercept and slope, which typically map to the first and second components respectively).

3) The number of patients should be clarified, and the exclusion of subjects in some analyses should be always justified.

We have clarified the number of patients and controls in the abstract. We also have stated on p. 5 the reason why four participants were excluded from sleep EEG analysis (due to persistent and severe line noise artifacts), and refer to a more detailed explanation in the Methods section. We have provided extended description of the exclusion criteria for wake ERP analysis on p. 42 in the Methods section.

4) The manuscript should be somehow simplified, especially the supplementary information. Many supplementary figures do not add helpful information.

We have removed nonessential supplementary figures – original Sup. Figures 3, 6-9. We note that, in response to reviewer requests, other new supplementary figures have been included in the revision.

5) All sleep analyses were performed on N2 epochs. Analyzing N2/N3 combined and/or N2 and N3 separately would likely provide a better characterization of these sleep parameters. Furthermore, because the schizophrenia group had less stage 2 sleep than the control group, the same number of epochs should be used in the analyses of both groups.

The duration of N2 stage was in fact not different between the groups. While there were marginally significant differences in N2 proportion in the whole sample (p=0.048), there were no significant group differences in N2 proportion in the analytic sample (i.e. after the exclusion of 4 studies due to noisy data) used for EEG analysis (p=0.14). This information is provided on p. 5 of the manuscript. In addition, the number of epochs used for the analyses of sleep microarchitecture also did not differ between SCZ and CTR (p=0.16). This is stated on p. 38 in the Methods section.

Our choice of performing all analyses using N2 epochs were motivated by our primary focus on spindle deficit in SCZ extensively reported for the N2 stage and that such effects were attenuated during N3 (Lai et al., 2022). As requested, however, we have now repeated the core analyses using N2 and N3 data combined. The results were broadly similar (except for SO density which was not significant after the inclusion of N3 data). We now mention this in the revised manuscript (p.10)

6) Some of the limitations should be clearly stated, for example that no adaption night was included in the study.

We have added this as a limitation on p. 32.

Reviewer #2 (Recommendations for the authors):In addition to the concerns raised in the public review (see weaknesses), a couple of additional points:The authors should clarify why they did not perform an initial analysis targeting the entire spindle range, but instead decided to focus on slow spindles and fast spindles. Indeed, in their previous sleep study (Purcell et al., 2017, ref. 14) the same wavelet-based algorithm employed here was used to initially target the whole spindle activity range and was centered at 13.5 Hz. Also, the choice of centering the analysis for slow spindles at 11 Hz and for fast spindles at 15 Hz is somewhat surprising, given that these frequency bins are usually considered the lowest and highest values respectively of the spindle activity. Furthermore, it appears that all sleep analyses were performed on N2 epochs. However, analyzing N2/N3 combined and/or N2 and N3 separately would likely provide a better characterization of these sleep parameters.

We chose target frequencies of 11 and 15 Hz to minimize the overlap between the detected spindles and maximize their differences in topography (frontal vs central) and coupling with SOs. We have stated it in the methods section (p.37) and added a supplementary figure that illustrates spindle properties for spindles detected using a range of target frequencies (10-16 Hz).

To further illustrate, the schematic plot shows amplitudes for the 11 Hz and 15 Hz wavelets, as well as a 13.5 Hz one (all with 7 cycles). As noted above, 11 and 15 Hz were selected to minimize overlap between "fast" and "slow" spindle definitions, but at the same time provide adequate coverage across the entire σ range. Practically, we have observed in this and other samples, that with a sufficiently broad wavelet (frequency domain), results from, e.g. F_C_ = 14 Hz will be very similar to those with F_C_ = 15 Hz.

Upon the reviewer’s request, we now also report group differences for spindles detected using a target frequency of 13.5 Hz (see the bottom row in Figure 2 —figure supplement 3). Those are broadly similar to the results we report for SS and FS (e.g. decreased density, amplitude), although (as expected) some alterations specific to SS or FS are diminished or not present when considering this center range (e.g. ISA, duration, chirp). Given the substantial body of evidence that points to heterogeneity in spindle activity that can be indexed by spindle frequency, we elected to present what we still feel are the more informative, two-class analyses as primary.

We now include a new Supplementary Figure to show the impact of different F_C_ choices (from 10 to 16 Hz) on key metrics and results. The frequency-dependent patterns of, e.g. topography and SO coupling support our decision to focus on the two classes, although precise 'boundary' frequencies are likely overlapping.

In previous work, the authors emphasize the role of sleep spindles in relation to cognitive function. It is therefore unclear why in the present study they did not examine the association of sleep metrics with cognitive parameters.

We strongly agree that investigating links between cognition and sleep neurophysiology in SCZ is a fruitful direction. However, acknowledging other reviewer comments about the complexity and abundance of analyses (often performed in multiple samples) already present in this manuscript, we respectfully note that this is beyond the scope of the current manuscript.

Reviewer #3 (Recommendations for the authors):1) Why were 4 subjects removed for the primary analytic sample (page 7)?

We removed 4 subjects due to persistent line noise artifacts. In the revised manuscript it is stated on p. 5 where we first mention it as well as in the Methods section on p. 38.

2) The sample size in the abstract is a bit misleading. Please specific the number of schizophrenia participants and controls instead of writing 130 schizophrenia patients and control.

We have changed ‘In 130 participants’ to ‘In 72 schizophrenia (SCZ) patients and 58 controls’ in the abstract.

3) Figure 1: The authors only show topomaps at 3.25 Hz and 13.25 Hz in the topomaps because these frequencies show the maximal difference between the groups, but a difference at single frequency bin is not biologically meaningful and it is misleading. Please stick to the convention of showing differences between the groups across electrodes (topomap) for a frequency band. Same applies for the scatterplots.

Please see our response (also provided to the similar comment in the Overall Summary).

With regard to single frequency bin analyses, we would like to stress that our use of Welch's method significantly reduces the variance of spectral power estimates: power for each 30-second epoch is averaged over 14 overlapping 4-second windows (with a frequency resolution of 0.25 Hz per bin), and for a stage, power is again averaged over dozens or hundreds of epochs. As such, individually testing frequency bins across a full range, and using an approach to multiple testing that allows for highly correlated predictors (i.e. our use of permutation-based empirical significance estimates) is warranted.

Nonetheless, we agree that summarizing by classical frequency band is also a common approach to present findings in spectral power, and so we have included those analyses. Case/control differences in spectral power for classical frequency bands are now given in Figure 1 —figure supplement 1.

While bin-based and band-based results are congruent for σ band/range frequencies, our δ/theta (2-6 Hz) results are not well captured by classical band definitions. Author response image 1 clearly illustrates a physiological group difference in NREM power that does not respect the traditional (useful but ultimately arbitrary) convention of a precise and binary delineation of activity at 4 Hz. When effects do fall along classically-defined lines (e.g. σ), our bin-based analysis will also show this; when they do not (e.g. the 2-6 Hz result), a band-based approach runs the risk of false negatives. To avoid such misrepresentation, we kept the original way of reporting the power findings in the main text.

We would also like to further clarify that our analysis of spectral power was performed across all frequency bins from 0.5 to 25 Hz. We show raw data plots for the most significant metrics only, as it is not feasible to show plots for all metrics and all channels: as we employed rigorous multiple-test correction, this approach does not unduly capitalize on chance.

Finally, we note that the principal spectral component (PSC) approach also provides a complementary mode of analysis that does not rely on individual frequency bins. However, unlike band-based approaches, PSC forms linear combinations across all frequency bins (and channels) in a data-driven manner (which also implicitly indexes other parameterizations of EEG power spectra not reflected by band-based analysis, e.g. of spectral intercept and slope, which typically map to the first and second components respectively).

4) The authors write: With respect to SO phase, both SS and FS tended to occur earlier in SCZ, albeit this effect was only observed at a few parietal channels (P5 and P7 for SS, max e.s.=-1.07 SD and P4, P6 for FS, max e.s.=-0.65 SD)." Only one channel is highlighted in Figure 3 for SS and one for FS. I would not draw conclusions based on the results of one channel.

We have revised the Figure 3 and it follows a similar representation of the data and results to other figures. Consistent with the rest of the manuscript, the "raw data plots" only highlight the exemplar association from a given analysis, to give the reader a concrete view of the underlying distributions within and between groups. All statistical analyses were corrected for multiple testing, and so showing an exemplar plot is no different from reporting the maximal effect sizes, etc, at those channels.

Although the original manuscript highlighted that (in contrast to most other results) only a few parietal channels were associated, we've made our interpretation clearer by adding the qualifying phrase (p 12):

**“**With respect to SO phase, both SS and FS tended to occur earlier in SCZ, albeit this effect was only observed at a few parietal channels therefore precluding any strong conclusions (P5 and P7 for SS, max e.s.=-1.07 SD and P4, P6 for FS, max e.s.=-0.65 SD).**”**

5) What does chirp tell us about the neurobiology of the disorder? An additional way to characterize spindles doesn't necessarily tell us much unless the cellular/molecular mechanisms have been clarified.

Although spindle chirp is a relatively novel metric, it has been shown that such intra-spindle frequency deceleration is a basic property of both slow and fast spindles (Andrillon et al., 2011; Schönwald et al., 2011). While the neurobiology of chirp is yet to be studied extensively, there are some clues to the function of this spindle property. For example, TRN neurons gradually hyperpolarize and fire with fewer action potentials upon repeated bursting which may underlie the chirp during the “waning” of spindles (Barthó et al., 2014). Therefore, chirp could reflect intrinsic biophysical properties of TRN and thalamocortical neurons during this phase. Our findings of altered chirp in SCZ should motivate future studies to confirm and investigate the precise properties of neural mechanisms behind spindle chirp.

We have added this in the discussion on p 30.

6) For the PSI why are data presented frequency by frequency instead of averaging into bands? Again, showing the statistics for the most significant single frequency bin is misleading. I'm unsure how to read Figure 5 since from my understanding PSI is a measure between two channels but here each channel is shown with a single color. Perhaps I missed something here.

PSI is based on the change (slope) in phase lag across a range of frequencies: it is necessarily not a 'single frequency bin' metric. We used a 5-Hz frequency window, reporting only the center frequency, e.g. a (net) PSI at 3 Hz will be estimated with respect to a spanning +/- 2.5 Hz window (i.e. 0.5 to 5.5 Hz). This is described in the Method section on p. 40.

Similar to other connectivity metrics, PSI is indeed a measure between two channels. For the purpose of dimensionality reduction, we computed a channel-wise net PSI as the sum of normalized PSIs over all pairs involving that channel (following the original description in Nolte et al., 2008). Higher absolute net PSI indicates that the channel is functionally connected to other channels, either leading (positive values) or lagging (negative values) other channels, on average. This is explained in the result section on p. 15. Net PSI group differences were effectively mirrored in pair-wise PSI, as shown in Figure 5 bottom row.

7) There are too many supplementary figures which aren't all helpful. For example suppl Figure 3 is not necessary. I can't find a reference for Supplementary Figure 1. Not sure what Supplementary Figure 2 is showing and how it relates to the two groups. Please consider carefully if all these figures are necessary

We have removed nonessential supplementary figures (Sup. Figures 3, 6-9 in the manuscript version prior revision) and have also made sure that all remaining supplementary figures are referenced in the text.

8) In the abstract the authors write "The main sleep findings were replicated in a demographically distinct sample, and a joint model, based on multiple NREM components, predicted disease status in the replication cohort." Prediction is a strong word given that the AUC is 0.64.

Note that full model classification performance in the replication sample had an AUC of 0.73 (Figure 10): the AUC of 0.64 was from a reduced model based on only spindle density (included only to highlight the value of considering multiple parameters of sleep EEG). However, we agree that it is important not to gloss over the statistical versus clinical implications of a term such as 'prediction'. We have therefore added the phrase "…statistically predicted…" to the abstract, and we include a note in the Discussion on p. 31 that makes clear that these measures do not, in their current form, necessarily have utility for clinical decision making:

“…a model based on multiple domains better classified disease status in an independent sample, compared to spindle density alone. Although still distant from diagnostic application in clinical settings, such results suggest that alterations in these aspects of sleep electrophysiology convey additional information on the neurobiological underpinnings of SCZ and may be helpful in stratifying patients with SCZ or distinguishing them from other neuropsychiatric conditions…”

9) Because the schizophrenia group had less stage 2 sleep than the control group, the same number of epochs should be used in the analyses of both groups. Similarly, significantly more artifacts were removed from the Schizophrenia group which may introduce bias and reflect a problem with the automatic artifact removal. Any ideas why more artifacts were removed from the schizophrenia group?

There was no significant group difference (p=0.16) in epoch number for the primary analytic sample. There was a modest difference in the percentage of epochs removed during QC, with more epochs removed in controls compared to SCZ patients (this information is provided on p.36). The significance and the effect size of difference in percentage of removed epochs is much smaller with respect to the reported alterations in EEG microstructure (e.g. p=0.01, e.s.=-0.68 SD for removed epochs vs p=4×10^-6^, e.s.=-1.27 SD for FS density). In addition, when the percentage of removed epochs was added as a covariate to our model, we still observe the same group differences in SS and FS density.

10) The differential referencing of the GCRC data set will have a large impact on the data and you should consider excluding this data set if the goal is to directly compare these previously published data sets.

Due to polarity uncertainties, we did not use the GCRC sample for replication of any metrics related to SOs. Therefore, the GCRC cohort was only included in the replication analysis of spindle parameters and spectral power.

We agree that certain EEG metrics can be strongly influenced by choice of reference. In our data, the core spectral and spindle metrics considered in the GCRC did not show substantive differences between these two mastoid-based referencing schemes. As a sensitivity analysis, we computed both sets of metrics for F3, F4, C3 and C4 (i.e. the channels present for GCRC dataset) for the same 20 N2 epochs in the main GRINS sample. For all spindle metrics and for all channels, the correlation coefficient between linked and contralateral mastoid-based estimates was above 0.99. Further, there were no significant mean differences (all p>0.98). Similar results were observed for spectral power, with core results being near indistinguishable: Author response image 2 shows power spectral for an exemplar control subject, for linked (red) and contralateral (black) mastoid references.

Collectively, these sensitivity analyses support the inclusion of the GCRC dataset in the replication set (for spindle and spectral power metrics).

11) No adaptation night was included. This should be incorporated into the discussion as a limitation of the study.

This has been added as a limitation section p.32.

12) What was the interval between the PANSS and the sleep EEG?

The PANSS assessment was performed within a week from the date of a EEG recording. This has been clarified in the revised manuscript on p.35.

13) Was prior sleep controlled? The differences in slow oscillations could be due to differential sleep-wake history of patients versus controls.

As partially summarized above, patients with SCZ went to bed earlier, woke up earlier and spent longer TIB the night before EEG recording. To address the potential for confounding, we tested whether these parameters were (conditional on disease status) in fact associated with the SO metrics which varied between groups. We initially focused on the CTR group, who were not under hospital-imposed schedules and so showed greater variability in sleep-wake patterns, presumably reflecting differences in circadian rhythms to some degree. Overall, prior sleep metrics were unrelated to SO parameters, with the one exception of nominally (p<0.01) significant associations at only two parietal channels, between TIB and SO density (positive direction of effect). Critically, SCZ/CTR group differences remained significant for these two channels after adding prior TIB as a covariate, however. Therefore, we did not find broad evidence to suggest that SO alterations were due to different sleep wake-histories.

**Author response image 3. sa2fig3:** 

14) The authors write that a minority of recordings ended prior during sleep. How many were these and did they differ between the two groups.

In total, there were 39 recordings that ended in sleep ­– in 16 SCZ patients and 23 CTR participants (p=0.049 group difference in rates). Due to the protocol in the sleep clinic, studies were truncated if the participant slept too late. The TST of truncated studies was in fact longer than for non-truncated studies (b=3.1, p=0.002). Whether or not a study was truncated was not a significant predictor of spindle density at any channel (all p > 0.05). In addition, the number of channels with significant case/control differences in SS and FS remained the same when truncation status was added as a covariate. We now have added this information in the manuscript on p. 36.

Reviewer #4 (Recommendations for the authors):The two fundamental processes relevant for sleep regulation – sleep homeostasis and its circadian control were not considered in this study, which makes some of the central findings difficult to interpret.For example, as well known slow wave activity is much greater in the early night as compared to morning hours. Yet, this highly significant dynamics was not taken into account. Conversely, the incidence of sleep spindles normally increases across the night, showing largely an opposite trends to slow-wave activity. These dynamics may meaningfully affect some of the metrics studied here, especially the coupling between slow oscillations and spindles. Some of the data presented in the paper are difficult to understand unless further information is provided on the dynamics of sleep oscillations across the night. At the very least I would recommend analysing separately the first and the last sleep cycle, but the authors may have other ways of addressing this important omission.

We appreciate the reviewer’s call for a focus on the dynamics of sleep metrics across the night. We conducted additional analyses to investigate the dynamics of slow/fast spindle density and the rate of SO across sleep cycles. First, we selected all subjects with at least three sleep cycles (60 SCZ and 56 CTR) and estimated slow and fast spindle density and SO density during N2 epochs for each cycle (all subjects had at least 17 epochs of N2 per cycle).

The first row of Figure 2 —figure supplement 1 illustrates how density (averaged across all channels) changed across three cycles, separately for SCZ and CTR groups. The topoplots below the graphs illustrate whether there was a significant association (p<0.05) between cycle number and spindle or SO density, again separately for SCZ and CTR groups. The strongest cycle related effects were, as expected, observed for SO density, which decreased profoundly in both SCZ and CTR groups from the first cycle to the third. Spindle density showed more variable, but nominally significant, dynamics across NREM cycles. In controls, SS density decreased in frontal channels but increased in a few posterior channels, while in SCZ only temporal channels displayed a significant decrease in density overnight. Density of FS had a non-linear trajectory in both groups – decreasing initially followed by an increase. (Both these results are consistent with the trends we reported in Purcell et al., (2017) in population-based samples, Figure 3a.) SCZ/CTR group differences, when tested separately for each cycle, grew more widespread over time, although qualitatively similar effects were observed for all cycles. We have included this new analysis in the discussion on p. 8 and 10 with the Figure 2 —figure supplement 1.

Further, as well known, patients with schizophrenia can manifest major circadian abnormalities in their mood and sleep behaviour. This is especially important given that the distribution of sleep stages and sleep oscillations, and specifically sleep spindles, are under a strong circadian control (as we know from forced desynchrony studies). Therefore, it seems essential to provide further information about circadian characteristics of sleep in the population studied, and the potential role of relevant factors, such as light levels, which are, surprisingly, never mentioned in this study. Please provide data on the levels of light the patients and controls were exposed to during the experiments. As I presume there is no data available on DLMO or body temperature, it is difficult to assess whether and to what extent the potential dysfunction of the circadian clock affects the results presented here. I would recommend providing more data on the time of sleep onset and morning awakening, relative to the light-dark cycle. Furthermore, it would be essential to provide further information on the season (winter, spring, summer, fall) when the data were collected.

Please see our response below (also provided to the similar comment in the Overall Summary).

With regard to circadian rhythms, we would like to point out that our SCZ cohort was composed of hospital inpatients, whose circadian rhythms and sleep patterns were strongly influenced (and regularized) by hospital-imposed schedules. As such, this cohort is not ideally suited to address the influence of circadian disruption as observed in community-based settings: this has been noted as a limitation in the Discussion on p. 32.

On the other hand, it can also be seen as one strength of our study, inasmuch as profiles of NREM micro-architecture in SCZ are unlikely to reflect only the nonspecific effects of generally disrupted sleep. To confirm this analytically, we conducted additional analyses using data collected prior to the EEG night: i) a sleep habits questionnaire, ii) a two-week sleep journal and iii) bedtime and wake times the night before the EEG. Analyses across all three data types congruently indicated that SCZ patients tended to go to bed earlier than control participants (all p-values < 10^-10^). Estimated total time in bed (TIB) was also longer in SCZ patients compared to CTR (all p-values < 10^-8^). Importantly, these results matched the group differences we originally reported for the EEG night (earlier bedtime, p = 7x10^-18^; longer TIB, p = 1x10^-10^) suggesting that the EEG night was representative of the regular sleep patterns in our SCZ cohort, characterized by earlier bed times and longer sleep latencies, but otherwise exhibiting sleep and stage durations broadly comparable with the CTR group. We added this information on p. 5.

Earlier bedtime and earlier wake time (the latter is only significant based on the sleep journal and the night before EEG recording) indicated earlier chronotype in SCZ patients. Similar to our report, Martin et al., 2005 showed earlier bedtime and longer TIB in SCZ. However, a more recent study found high inter-subject variability in SCZ which suggested two subgroups of SCZ patients exist with either earlier or later bedtimes compared to controls (Wulff et al., 2012). As noted, given that our cohort is composed of inpatients whose sleep/wake timing were strongly determined by hospital schedules, this study was not designed to address their natural circadian rhythm.

Next, we directly investigated whether bedtime or TIB prior or during the EEG night were associated with sleep microarchitecture metrics altered in SCZ in channels with the largest effect size. Except the SO density at O1 which was positively associated with TIB of the night before in both groups (p=0.041 in CTR and p=0.042 in SCZ), none of the tested sleep microarchitecture parameters (i.e. slow and fast spindle density, amplitude, ISA, FS duration, and chirp; SO duration, and slope; slow spindle overlap with SOs, SO coupling angle at slow and fast spindle peaks) displayed significant associations with TIB or bedtime congruently in SCZ and CTR groups. Furthermore, despite the revealed significant association in SO density at O1 , adding TIB of the night before as covariate could not explain the observed SCZ/CTR group differences in SO rate (original p=5.5x10^-5^ vs p= 0.004 controlling for TIB of the night before ).

To address concerns regarding the seasonal/light exposure effects, we used the R package ‘suncalc’ to estimate an exact time of sunrise on the day of the EEG, finding no significant difference between SCZ and CTR groups (p=0.48). Regarding artificial light exposure during the procedures prior to sleep, all participants followed an identical protocol in the same hospital sleeping ward precluding systematic differences that could impact sleep.

References

Andrillon, T., Nir, Y., Staba, R. J., Ferrarelli, F., Cirelli, C., Tononi, G., and Fried, I. (2011). Sleep Spindles in Humans: Insights from Intracranial EEG and Unit Recordings. *The Journal of Neuroscience*, *31*(49), 17821–17834. https://doi.org/10.1523/JNEUROSCI.2604-11.2011

Barthó, P., Slézia, A., Mátyás, F., Faradzs-Zade, L., Ulbert, I., Harris, K. D., and Acsády, L. (2014). Ongoing Network State Controls the Length of Sleep Spindles via Inhibitory Activity. *Neuron*, *82*(6), 1367–1379. https://doi.org/10.1016/j.neuron.2014.04.046

Lai, M., Hegde, R., Kelly, S., Bannai, D., Lizano, P., Stickgold, R., Manoach, D. S., and Keshavan, M. (2022). Investigating sleep spindle density and schizophrenia: A meta-analysis. *Psychiatry Research*, *307*, 114265. https://doi.org/10.1016/j.psychres.2021.114265

Martin, J. L., Jeste, D. V., and Ancoli-Israel, S. (2005). Older schizophrenia patients have more disrupted sleep and circadian rhythms than age-matched comparison subjects. *Journal of Psychiatric Research*, *39*(3), 251–259. https://doi.org/10.1016/j.jpsychires.2004.08.011

Schönwald, S. V., Carvalho, D. Z., Dellagustin, G., de Santa-Helena, E. L., and Gerhardt, G. J. L. (2011). Quantifying chirp in sleep spindles. *Journal of Neuroscience Methods*, *197*(1), 158–164. https://doi.org/10.1016/j.jneumeth.2011.01.025

Wulff, K., Dijk, D.-J., Middleton, B., Foster, R. G., and Joyce, E. M. (2012). Sleep and circadian rhythm disruption in schizophrenia. *The British Journal of Psychiatry*, *200*(4), 308–316. https://doi.org/10.1192/bjp.bp.111.096321